# Spark Transformer: How Many FLOPs is a Token Worth?

## Abstract

This work introduces *Spark Transformer*, an architectural variant of the Transformer model that drastically reduces the FLOPs count while maintaining comparable quality and an identical parameter count. This reduction is achieved by introducing sparse activations in both the feedforward network (FFN) and the Attention mechanism. In the FFN, this sparsity engages only a subset of parameters for each input. In the Attention mechanism, it limits the number of tokens that each token attends to. We achieve this sparsity through *statistical top-$k$*, a lightweight approximate algorithm that is well-suited for accelerator hardware and minimizes training slowdown. Furthermore, Spark Transformer incorporates dedicated predictors to identify the activated entries. These predictors are formed by allocating a portion of the model's parameters and are trained jointly with the rest of the model. This approach distinguishes Spark Transformer from existing methods that introduce sparsity and predictors post-training, which often leads to increased training costs, additional model parameters, and complex modifications to the model architecture. Our Spark Transformer, pretrained using the Gemma 2 recipe, achieves competitive performance on standard benchmarks while exhibiting significant sparsity. Specifically, it utilizes only $8\%$ nonzeros in the FFN activation and attends to a maximum of 256 tokens. This results in a $3.1\times$ reduction in FLOPs, yielding a $1.70\times$ speedup for prefill and a $1.79\times$ speedup for decoding on a 16-core CPU VM.

## 1 Introduction

The machine learning landscape has witnessed a surge in large-scale Transformer models (Anil et al., 2023; Almazrouei et al., 2023; Dubey et al., 2024; Adler et al., 2024), pushing the boundaries of language understanding and generation. However, the pursuit of scale is often constrained not by limitations in model quality, but by the escalating computational costs (Sharir et al., 2020; Patterson et al., 2021) associated with increasing parameter counts (Kaplan et al., 2020). This challenge is further exacerbated by the development of models that handle increasingly long context inputs (Reid et al., 2024), where computational demands grow proportionally with context length.

*Sparse activation* is a popular approach for addressing the computational challenges posed by both large model size and long context length. To handle large models, sparse activation reduces costs by engaging only a small subset of model parameters for each input. This approach has gained significant interest following the discovery of an intriguing phenomenon: the feed-forward networks (FFNs) in classic Transformers like T5 (Raffel et al., 2020) and ViT (Dosovitskiy, 2020) exhibit *natural* activation sparsity (Zhang et al., 2022; Li et al., 2022). In other words, these models demonstrate sparsity without explicit enforcement. This inherent sparsity, reaching remarkable levels like 3% nonzeros in T5, presents an opportunity for substantial efficiency gains with minimal modifications to the model architecture.

Unfortunately, this natural sparsity is absent in the latest generation of models (Jiang et al., 2023; Gemma Team, 2024; Dubey et al., 2024), which have adopted *gated non-ReLU* activation functions (Dauphin et al., 2017). To re-introduce sparsity, recent work has explored extra pretraining steps, either by switching back to ReLU activations (Mirzadeh et al., 2023; Zhang et al., 2024) or by adding top-$k$ thresholding (Yerram et al., 2024; Song et al., 2024a;b). This is often combined with training a low-cost predictor to identify activated parameters, a crucial step for maximizing efficiency gains (Liu et al., 2023; Zeng et al., 2023; Song et al., 2023; Yerram et al., 2024). However,

Table 1: **FLOPs per token comparison: Spark Transformer vs. standard Transformer.** In a standard Transformer with model dimension $d_{\text{model}}$, we assume multi-head attention where the sum of head dimensions equals $d_{\text{model}}$, and an FFN with non-gated activation and width $d_{\text{ff}}$. Here, $n_{\text{ctx}}$ represents the context length for the target token. The computational cost is primarily determined by the FFN (assuming $d_{\text{ff}} \gg d_{\text{model}}$, which is typical) and the attention dot product (assuming a long context length). Spark Transformers introduce sparsity parameters, $k_{\text{ff}}$ and $k_{\text{attn}}$, to reduce FLOPs. Setting $k_{\text{ff}} = 8\% \times d_{\text{ff}}$ and $k_{\text{attn}} = 256$ achieves a $3.2\times$ FLOPs reduction in the FFN, a $4\times$ reduction in the attention dot product, and a $3.1\times$ reduction overall (assuming $n_{\text{ctx}} = 8k$) for Gemma-2B.

| Operation | FLOPs per Token[1] | |
|---|---|---|
| | Standard Transformer | Spark Transformer (Ours) |
| FFN | $2d_{\text{model}}d_{\text{ff}}$ | $0.5d_{\text{model}}d_{\text{ff}} + 1.5d_{\text{model}}k_{\text{ff}}$ |
| Attention dot product | $2d_{\text{model}}n_{\text{ctx}}$ | $0.5d_{\text{model}}n_{\text{ctx}} + 1.5d_{\text{model}}\min\{k_{\text{attn}}, n_{\text{ctx}}\}$ |
| Attention linear projection | $4d_{\text{model}}^2$ | $4d_{\text{model}}^2$ |

these approaches not only complicate the training procedure and incur extra training costs, but also introduce additional parameters (for the predictor) and have yet to demonstrate high sparsity levels without compromising model quality.

Sparse activation for attention, often called sparse attention, faces a similar challenge. Sparsity is used to efficiently handle long context input by limiting the number of tokens each token attends to. A straightforward approach is top-$k$ attention (Gupta et al., 2021), which applies a top-$k$ mask to the attention coefficients. This can be combined with a low-cost predictor to maximize efficiency (Ribar et al., 2023; Yang et al., 2024; Lee et al., 2024c). However, achieving high sparsity and a predictor without complicated procedure and sacrificing quality remains a challenge.

**Contributions.** This work introduces *Spark Transformer*, an architectural variant of Transformer that achieves both high activation sparsity and low-cost prediction in both the FFN and Attention mechanisms. Notably, Spark Transformer can be trained in a single stage without requiring separate post-processing and maintains quality without introducing additional parameters. This makes it a suitable drop-in replacement for standard Transformer models. Because the FFN and attention components dominate the computational cost in large Transformers with long contexts, Spark Transformer drastically reduces the overall FLOP count for decoding a token (see Table 1).

We pretrain a Spark Transformer using the Gemma-2 recipe (Gemma Team, 2024), resulting in a model we call Spark Gemma-2. Evaluation on standard benchmarks demonstrates that Spark Gemma-2 closely matches the quality of Gemma-2, even with a high degree of sparsity: only 8% activated entries in the FFN and a maximum of 256 attended tokens in attention (see Table 2). This sparsity leads to a $3.1\times$ reduction in overall FLOPs compared to Gemma-2. Using this sparsity, we evaluate model efficiency with `gemma.cpp` (Google Gemma.cpp, 2024), a C++ inference engine optimized for serving Gemma models on CPUs, and observe a speedup of up to $1.79\times$ (see Figure 3). Notably, on a readily available 4-core cloud VM, Spark Gemma-2 achieves a decoding speed of 86 ms per token, surpassing typical human reading speed (Brysbaert, 2019). This increased efficiency enables wider access to high-quality models for users with limited access to high-FLOP devices, such as GPUs and TPUs.

Spark Transformer leverages the interpretation of both FFN and attention as key-value lookup tables (Geva et al., 2021) to provide a unified solution for achieving sparsity and prediction. Specifically, the predictor is obtained by repurposing a subset of the dimensions of the query and key vectors to produce an importance score for each key-value pair (see Section 3). Top-$k$ thresholding is applied to these scores to identify the activated keys. In particular, standard top-$k$ thresholding requires performing a sorting, which is inefficient particularly on training accelerators. To address this, we introduce *statistical top-$k$*, a linear complexity algorithm for approximate nearest neighbor search (see Section 2) based on fitting a Gaussian distribution to the activation entries and estimating a threshold that yields the top entries. While ideas similar to statistical top-$k$ have been used (Shi et al., 2019; M Abdelmoniem et al., 2021) for the problem of distributed training (Lin et al., 2018), we are the first to introduce, adapt, and verify its effectiveness for activation sparsity.

---

[1]Please refer to Section 3.1 and Section 3.2 for the calculation of FLOPs for FFN and Attention, respectively. We omit non-leading-order terms (e.g., those arising from embedding, normalization, and nonlinear layers) and exclude the number of layers as a common multiplier.

## 2 STATISTICAL TOP-K

This section introduces $\text{Statistical-Top}_k$, an approximate algorithm for obtaining the $k$ largest entries of an input vector. Recall that the *soft-thresholding operator* is defined for an arbitrary vector $\boldsymbol{x} \in \mathbb{R}^d$ and a scalar threshold $\theta \in \mathbb{R}$ as

$$\text{Soft-Threshold}(\boldsymbol{x}, \theta) \overset{\text{def}}{=} \max\{\boldsymbol{x} - \theta \cdot \mathbf{1},\, \mathbf{0}\} \in \mathbb{R}^d, \tag{1}$$

where $\mathbf{1}$ and $\mathbf{0}$ are $d$-dimensional vectors with all entries equal to 1 and 0, respectively. The soft-thresholding operator shifts each entry of $\boldsymbol{x}$ to the left by $\theta$ and then thresholds the result at zero.

We define $\text{Statistical-Top}_k$ as the following mapping from $\mathbb{R}^d$ to $\mathbb{R}^d$:

$$\text{Statistical-Top}_k(\boldsymbol{x}) \overset{\text{def}}{=} \text{Soft-Threshold}(\boldsymbol{x}, \theta(\boldsymbol{x}, k)), \text{ where } \theta(\boldsymbol{x}, k) \overset{\text{def}}{=} \text{mean}(\boldsymbol{x}) + \text{std}(\boldsymbol{x}) \cdot Q(1 - \tfrac{k}{d}) \tag{2}$$

Here, $\text{mean}(\boldsymbol{x}) \overset{\text{def}}{=} \frac{1}{d} \sum_{i=1}^d x_i$ and $\text{std}(\boldsymbol{x}) \overset{\text{def}}{=} \sqrt{\frac{1}{d-1} \sum_{i=1}^d (x_i - \text{mean}(\boldsymbol{x}))^2}$ compute the sample mean and standard deviation of the entries of $\boldsymbol{x}$, respectively, and $Q(\cdot)$ is the quantile function (i.e., inverse of the cumulative distribution function) of the standard Gaussian distribution.

$\text{Statistical-Top}_k$ in Eq. (2) operates by first computing a threshold $\theta(\boldsymbol{x}, k)$ such that approximately $k$ entries of $\boldsymbol{x}$ exceed it, and then applying the soft-thresholding operator with this threshold to $\boldsymbol{x}$ to obtain a sparse output. We discuss these two components in the next two subsections.

### 2.1 THRESHOLD ESTIMATION

The threshold $\theta(\boldsymbol{x}, k)$ in Eq. (2) is designed such that, if the entries of $\boldsymbol{x}$ are drawn from a Gaussian distribution, approximately $k$ out of the $d$ entries will exceed this threshold. To understand this, let $\mu$ and $\sigma$ denote the mean and standard deviation of the underlying Gaussian distribution. Its quantile function is given by $\mu + \sigma \cdot Q(p)$ for $p \in (0, 1)$. Consequently, due to the properties of quantile functions, we expect roughly $p \cdot d$ entries of $\boldsymbol{x}$ to exceed $\mu + \sigma \cdot Q(1 - p)$. In practice, since $\mu$ and $\sigma$ are unknown, they are replaced with the sample mean $\text{mean}(\boldsymbol{x})$ and the sample standard deviation $\text{std}(\boldsymbol{x})$, respectively.

The following theorem formalizes this argument.

**Theorem 1.** *Let $\boldsymbol{x} \in \mathbb{R}^d$ be a vector with entries drawn i.i.d. from $\mathcal{N}(\mu, \sigma^2)$. For any $1 \le k \le d-1$, let $\theta(\boldsymbol{x}, k)$ be a scalar defined in Eq. (2). Take any $\delta \in (0, 1)$ and assume $d \ge \max\{2, \log \frac{6}{\delta}\}$. With a probability of at least $1 - \delta$, the number of entries of $\boldsymbol{x}$ that are greater than $\theta(\boldsymbol{x}, k)$, i.e., $\mathbf{card}\left(\{i \in [d] \mid x_i > \theta(\boldsymbol{x}, k)\}\right)$, satisfies*

$$\frac{|\mathbf{card}\left(\{i \in [d] \mid x_i > \theta(\boldsymbol{x}, k)\}\right) - k|}{d} \le 4\sqrt{\frac{\log \frac{6}{\delta}}{d}} \left(1 + \sqrt{-2 \log \min\left\{\frac{k}{d}, 1 - \frac{k}{d}\right\}}\right).$$

Theorem 1 provides a relative error bound between $k$ and the true number of entries of $\boldsymbol{x}$ that exceed $k$. This bound is maximized when $k = 1$ or $k = d - 1$. Consequently, the worst-case bound is $O\left(\sqrt{\frac{\log d \cdot \log \frac{1}{\delta}}{d}}\right)$ which vanishes as $d$ increases. Notably, the error bound becomes $O\left(\sqrt{\frac{\log \frac{1}{\delta}}{d}}\right)$ when $k = \Theta(d)$, demonstrating even faster convergence.

**Computation cost.** The computation of the threshold $\theta(\mathbf{x}, k)$ is highly efficient, requiring only $2d$ FLOPs to compute the mean and standard deviation of the samples. This contrasts sharply with a naive sorting-based approach, which has $O(d \log d)$ complexity.

While the Gaussian quantile function $Q(\cdot)$ lacks a closed-form solution, high-precision piecewise approximation algorithms with constant complexity are available in standard software packages like SciPy (Virtanen et al., 2020), readily applicable to our needs.

### 2.2 SPARSIFICATION

Given the threshold $\theta(\boldsymbol{x}, k)$, a straightforward approach to obtain a sparse vector is to set all entries of $\boldsymbol{x}$ below the threshold to zero, preserving the remaining values. This operator, sometimes referred to as *hard thresholding* (Blumensath & Davies, 2008), suffers from discontinuity, potentially hindering its suitability for gradient-descent-based training.

To address this, $\text{Statistical-Top}_k$ employs the soft-thresholding operator defined in Eq. (1) (Beck & Teboulle, 2009). This operator first shrinks all entries of $\boldsymbol{x}$ by the threshold $\theta(\boldsymbol{x}, k)$ and then sets all entries below 0 to 0. Soft thresholding offers the advantages of being continuous and differentiable almost everywhere (except when entries of $\boldsymbol{x}$ coincide with $\theta(\boldsymbol{x}, k)$).

For complete differentiability, one can utilize a smoothing function like the Huber loss (Huber, 1992), defined element-wise on an input $\boldsymbol{x}$ as:

$$\text{Huber}(x; \delta) \stackrel{\text{def}}{=} \begin{cases} \frac{1}{2}x^2 & \text{for } |x| < \delta, \\ \delta \cdot (|x| - \frac{1}{2}\delta) & \text{otherwise.} \end{cases} \tag{3}$$

The following theorem establishes the continuous differentiability of the mapping $\boldsymbol{x} \mapsto \text{Huber}(\text{Statistical-Top}_k(\boldsymbol{x}); \delta)/\delta$:

**Theorem 2.** *For any $\delta > 0$, the function $\mathbb{R}^d \to \mathbb{R}^d$ defined as*

$$\text{Huber}(\text{Statistical-Top}_k(\boldsymbol{x}); \delta) \, / \, \delta \tag{4}$$

*is continuously differentiable.*

Note that Eq. (4) converges to $\text{Statistical-Top}_k(\boldsymbol{x})$ as $\delta \to 0$, since $\text{Huber}(x; \delta)/\delta \to |x|$ and $\text{Statistical-Top}_k(\boldsymbol{x})$ is always non-negative. In practice, however, we find that using a non-zero $\delta$ does not improve model quality, and therefore we set $\delta = 0$ for simplicity.

Finally, soft thresholding admits a variational form (see, e.g., Parikh et al. (2014)):

$$\text{Soft-Threshold}(\boldsymbol{x}, \theta) = \underset{\boldsymbol{z} \geq \boldsymbol{0}}{\arg\min} \, \theta\|\boldsymbol{z}\|_1 + \frac{1}{2}\|\boldsymbol{x} - \boldsymbol{z}\|_2^2. \tag{5}$$

This formulation seeks a vector $\boldsymbol{z}$ that minimizes both its squared $\ell_2$ distance to the input $\boldsymbol{x}$ and its $\ell_1$ norm, with the threshold $\theta$ balancing these terms. Given the sparsity-promoting nature of the $\ell_1$ norm, soft thresholding effectively finds a sparse approximation of the input $\boldsymbol{x}$.

### 2.3 Comparison with related top-$k$ operators

The variational form in Eq. (5) also reveals connections of $\text{Statistical-Top}_k$ with other top-$k$ algorithms in the literature. Specifically, Lei et al. (2023) defines a *soft top-$k$* as

$$\underset{\boldsymbol{z}}{\arg\min} \quad -\theta \cdot H(\boldsymbol{z}) - \langle \boldsymbol{z}, \boldsymbol{x} \rangle, \quad \text{s.t. } \boldsymbol{z}^\top \boldsymbol{1} = k, \; \boldsymbol{0} \leq \boldsymbol{z} \leq \boldsymbol{1}, \tag{6}$$

where $H(\boldsymbol{z})$ is the entropy function. Another work (Lou et al., 2024) defines the *SparseK* operator

$$\underset{\boldsymbol{z}}{\arg\min} \quad -H^G(\boldsymbol{z}) - \langle \boldsymbol{z}, \boldsymbol{x} \rangle, \quad \text{s.t. } \boldsymbol{z}^\top \boldsymbol{1} = k, \; \boldsymbol{0} \leq \boldsymbol{z} \leq \boldsymbol{1}, \tag{7}$$

where $H^G(\boldsymbol{z})$ is the generalized Gini entropy.

$\text{Statistical-Top}_k$ in the form of Eq. (5), as well as Eq. (6) and Eq. (7), can all be interpreted as finding an output that is *close* to the input subject to a sparsifying regularization. Their major difference lies in the choice of the sparse regularization. That is, $\text{Statistical-Top}_k$ uses $\ell_1$, whereas soft top-$k$ and SparseK uses entropy and Gini entropy, respectively. The choice of $\ell_1$ makes $\text{Statistical-Top}_k$ superior in that it has a closed form solution provided by soft-thresholding, which only requires $d$ FLOPs. In contrast, soft top-$k$ and SparseK both do not have closed form solutions and require an iterative algorithm with a FLOP count dependent on the number of iterations. In addition, there is no guarantee that soft top-$k$ and SparseK can obtain (approximately) $k$ nonzero entries as output.

## 3 Spark Transformer

This section describes Spark FFN and Spark Attention, the two components of Spark Transformer.

### 3.1 Spark FFN

FFNs in a standard Transformer are two-layer multi-layer perceptrons that map an input token $\boldsymbol{q} \in \mathbb{R}^{d_{\text{model}}}$ to an output

$$\text{FFN}(\boldsymbol{q}; \boldsymbol{K}, \boldsymbol{V}) \stackrel{\text{def}}{=} \boldsymbol{V} \cdot \sigma\left(\boldsymbol{K}^\top \boldsymbol{q}\right) \in \mathbb{R}^{d_{\text{model}}}. \tag{8}$$

In above, $\{\boldsymbol{K}, \boldsymbol{V}\} \subseteq \mathbb{R}^{d_{\text{model}} \times d_{\text{ff}}}$ are trainable model parameters, and $\sigma()$ is a nonlinear activation function. We ignore the dependency on layer index to simplify the notations.

Each of the matrix multiplication in Eq. (8) has $d_{\text{model}} \cdot d_{\text{ff}}$ FLOPs hence overall the computation cost is $2d_{\text{model}} \cdot d_{\text{ff}}$. However, previous work shows that when $\sigma()$ is ReLU, the activation map $\sigma(\boldsymbol{K}^\top \boldsymbol{q})$ is very sparse after model training. The sparsity can be used trivially to reduce the computation costs in the calculation of its product with the second layer weight matrix $\boldsymbol{V}$ (Li et al., 2022), reducing the overall FLOPs count of FFN to $d_{\text{model}} \cdot (d_{\text{ff}} + k)$, where $k \ll d_{\text{ff}}$ is the number of nonzero entries in the activation. Note that the sparsity cannot be used to reduce the computation costs associated with $\boldsymbol{K}$, which constitute half of the total FLOPs in FFN.

In order to reduce FLOPs count in the first layer of FFN as well, we introduce Spark FFN as follows:

$$\text{Spark-FFN}(\boldsymbol{q}; \boldsymbol{K}, \boldsymbol{V}, k, r) \overset{\text{def}}{=} \boldsymbol{V} \cdot \big(\sigma\big(\text{Statistical-Top}_k(\boldsymbol{K}^\top \boldsymbol{P} \boldsymbol{q})\big) \odot \big(\boldsymbol{K}^\top (\boldsymbol{I} - \boldsymbol{P})\boldsymbol{q}\big)\big) \in \mathbb{R}^{d_{\text{model}}}. \tag{9}$$

In above, $\{\boldsymbol{K}, \boldsymbol{V}\} \subseteq \mathbb{R}^{d_{\text{model}} \times d_{\text{ff}}}$ are trainable parameters as in standard FFNs, and the activation $\sigma()$ is taken to be GELU (Hendrycks & Gimpel, 2016) following Gemma. The $\text{Statistical-Top}_k$, defined in Eq. (2), is introduced for obtaining sparsity, with $k$ being a hyper-parameter specifying the sparsity level. Finally, $\boldsymbol{P}$ is a fixed matrix $\boldsymbol{P} \overset{\text{def}}{=} \boldsymbol{1}_r \bigoplus \boldsymbol{0}_{d_{\text{model}}-r} \in \mathbb{R}^{d_{\text{model}} \times d_{\text{model}}}$ where $\bigoplus$ is the direct sum operator and $r$ is a hyper-parameter. It is introduced so that the term $\boldsymbol{K}^\top \boldsymbol{P} \boldsymbol{q}$ serves as a low-rank predictor of the location of the nonzero entries, which allows us to obtain efficiency benefits in computing $\boldsymbol{K}^\top (\boldsymbol{I} - \boldsymbol{P})$ and the multiplication with $\boldsymbol{V}$. This is discussed in detail below.

**FLOPs per Token.** Naive implementation of the Spark-FFN has the same number of FLOPs as the vanilla FFN in Eq. (8), i.e.,

$$r \cdot d_{\text{ff}} + (d_{\text{model}} - r) \cdot d_{\text{ff}} + d_{\text{model}} \cdot d_{\text{ff}} = 2d_{\text{model}} \times d_{\text{ff}} \tag{10}$$

where the three terms are from $\boldsymbol{K}^\top \boldsymbol{P} \boldsymbol{q}$, $\boldsymbol{K}^\top (\boldsymbol{I} - \boldsymbol{P})\boldsymbol{q}$, and the multiplication with $\boldsymbol{V}$, respectively. In Spark-FFN, one may first compute the term $\boldsymbol{K}^\top \boldsymbol{P} \boldsymbol{q}$ as a low-rank predictor. After passing its output through $\text{Statistical-Top}_k$, which selects approximately the $k$ most important entries, followed by the activation function $\sigma()$, we obtain a sparse output. Importantly, after obtaining the sparse output there is no need to perform the full computation of the other two matrix multiplications in Eq. (9), i.e., $\boldsymbol{K}^\top (\boldsymbol{I} - \boldsymbol{P})\boldsymbol{q}$ and the multiplication with $\boldsymbol{V}$. Instead, one can perform a sparse matrix multiplication with a drastically reduced FLOPs count:

$$r \cdot d_{\text{ff}} + (d_{\text{model}} - r) \cdot k + d_{\text{model}} \cdot k = (d_{\text{ff}} - k) \cdot r + 2d_{\text{model}} \cdot k, \tag{11}$$

which is an increasing function of $r$. In other words, $r$ controls the computation cost. We provide ablation study in the section to show that the best model quality is obtained when $r \approx \frac{d_{\text{model}}}{2}$. In this case, the total FLOP count of Spark FFN is approximately $0.5 \cdot d_{\text{model}} \cdot d_{\text{ff}} + 1.5 \cdot d_{\text{model}} \cdot k$, which is a 4-times reduction from Eq. (10) when $k$ is very small.

**Relation to gated activation.** Many of the most recent Transformers, including Gemma 2, use a variant of the standard FFN in Eq. (8) where the activation function is replaced with a gated one:

$$\text{Gated-FFN}(\boldsymbol{q}; \boldsymbol{K}_1, \boldsymbol{K}_2, \boldsymbol{V}) = \boldsymbol{V} \cdot \big(\sigma\big(\boldsymbol{K}_1^\top \boldsymbol{q}\big) \odot \big(\boldsymbol{K}_2^\top \boldsymbol{q}\big)\big). \tag{12}$$

Note that when compared with the FFN in Eq. (8) for quality studies, $d'_{\text{ff}}$ is usually taken to be $2/3 \cdot d_{\text{ff}}$ to be iso-parameter count (Shazeer, 2020).

Our Spark FFN in Eq. (9) bears some resemblance to Gated FFN in that both have two linear maps in the first layer and one in the second layer. The difference lies in that 1) Spark FFN adds a statistical top-$k$ to obtain sparsity, and 2) the input to the first layers of Spark FFN are obtained from splitting the dimensions of the input. The latter change has the benefit that it offers a convenient means of controlling the number of FLOPs from tuning the choice of $r$ (see Eq. (11)).

## 3.2 SPARK ATTENTION

In a standard multi-head attention layer, an input $\boldsymbol{x} \in \mathbb{R}^{d_{\text{model}}}$ is mapped to a query, a key, and a value vector of dimension $d_{\text{attn}}$ as $\boldsymbol{q}^{(i)} = \boldsymbol{W}_Q^{(i)} \boldsymbol{x} \in \mathbb{R}^{d_{\text{attn}}}, \boldsymbol{k}^{(i)} = \boldsymbol{W}_K^{(i)} \boldsymbol{x} \in \mathbb{R}^{d_{\text{attn}}}, \boldsymbol{v}^{(i)} = \boldsymbol{W}_V^{(i)} \boldsymbol{x} \in \mathbb{R}^{d_{\text{attn}}}$ for each head $i$. Here, $\{\boldsymbol{W}_Q^{(i)}, \boldsymbol{W}_K^{(i)}, \boldsymbol{W}_V^{(i)}\} \subseteq \mathbb{R}^{d_{\text{attn}} \times d_{\text{model}}}$ are trainable weights.

Collecting all the key and value vectors in the context of $\boldsymbol{x}$ into $\boldsymbol{K}^{(i)} = [\boldsymbol{k}_1^{(i)}, \ldots, \boldsymbol{k}_{n_{\text{ctx}}}^{(i)}] \in \mathbb{R}^{d_{\text{attn}} \times n_{\text{ctx}}}$ and $\boldsymbol{V}^{(i)} = [\boldsymbol{v}_1^{(i)}, \ldots, \boldsymbol{v}_{n_{\text{ctx}}}^{(i)}] \in \mathbb{R}^{d_{\text{attn}} \times n_{\text{ctx}}}$, attention conducts the following computation:

$$\text{Attention}(\boldsymbol{q}; \boldsymbol{K}, \boldsymbol{V}) \overset{\text{def}}{=} \boldsymbol{V} \cdot \text{softmax}\big(\boldsymbol{K}^\top \boldsymbol{q}\big) \in \mathbb{R}^{d_{\text{attn}}}, \tag{13}$$

where we omit the dependency on $i$ for simplicity. Note that computation cost associated with Eq. (13) is $2d_{\text{attn}} \cdot n_{\text{ctx}}$ for each head. Finally, output from all heads are concatenated followed by a linear map to project to $d_{\text{model}}$.

Note that Eq. (13) has the same form as FFN in Eq. (8) except for the choice of nonlinearity. Hence, following a similar strategy in obtaining Spark FFN, here we present Spark Attention as

$$\text{Spark-Attention}(\boldsymbol{q}; \boldsymbol{K}, \boldsymbol{V}, k, r) \overset{\text{def}}{=}$$
$$\boldsymbol{V} \cdot \left( \text{softmax}\left( \text{Statistical-Top}_k^{(-\infty)}(\boldsymbol{K}^\top \boldsymbol{P}\boldsymbol{q}) \right) \odot \text{softplus}\left( \boldsymbol{K}^\top (\boldsymbol{I} - \boldsymbol{P})\boldsymbol{q} \right) \right) \quad (14)$$

In above, $\boldsymbol{P} \overset{\text{def}}{=} \boldsymbol{1}_r \bigoplus \boldsymbol{0}_{d_{\text{attn}}-r} \in \mathbb{R}^{d_{\text{attn}} \times d_{\text{attn}}}$ is a fixed matrix. $\text{Statistical-Top}_k^{(-\infty)}$ is a slight variant of Eq. (2) where the entries below the threshold $\theta(\boldsymbol{x}, k)$ are set to $-\infty$ instead of 0, so that such entries become zero after passing through softmax. Specifically,

$$[\text{Statistical-Top}_k^{(-\infty)}(\boldsymbol{x})]_i \overset{\text{def}}{=} \begin{cases} \boldsymbol{x}_i - \theta(\boldsymbol{x}, k) & \text{if } \boldsymbol{x}_i > \theta(\boldsymbol{x}, k), \\ -\infty & \text{otherwise.} \end{cases} \quad (15)$$

Finally, a $\text{softplus}$ nonlinearity defined as the entrywise softplus function, i.e., $\log(1 + \exp(x))$, is applied to the term $\boldsymbol{K}^\top (\boldsymbol{I} - \boldsymbol{P})\boldsymbol{q}$ as this is empirically observed to offer quality benefits.

**FLOPs per Token.** With a naive implementation the number of FLOPs in Eq. (14) is given by $2d_{\text{attn}} \cdot n_{\text{ctx}}$, which is the same as the FLOPs for Eq. (13). However, by noting that the output of the softmax is expected to be sparse with approximately $k$ nonzero entries, the computation costs associated with $\boldsymbol{K}^\top (\boldsymbol{I} - \boldsymbol{P})\boldsymbol{q}$ and in the multiplication with $\boldsymbol{V}$ can be drastically reduced. In particular, if we take $r = \frac{d_{\text{attn}}}{2}$ then the FLOPs per token becomes

$$0.5d_{\text{model}}n_{\text{ctx}} + 1.5d_{\text{model}}\min\{k_{\text{attn}}, n_{\text{ctx}}\}, \quad (16)$$

which is nearly a $4\times$ reduction when $k_{\text{attn}}$ is much smaller than $n_{\text{ctx}}$.

## 4 EXPERIMENTS

In this section, we present an experimental evaluation of Spark Transformer using the Gemma-2 2B model. Gemma-2 2B is a decoder-only Transformer with 2 billion parameters, pretrained on 2 trillion tokens of primarily English text data (see Gemma Team (2024) for details). To evaluate Spark Transformer, we train a *Spark Gemma-2 2B* model by substituting the standard FFN and Attention in Gemma-2 2B with their Spark Transformer counterparts (Spark FFN and Spark Attention, respectively). This Spark Gemma-2 2B model is trained using the same procedure and data as the original Gemma-2 2B model.

**Implementation details.** Gemma-2 uses a model dimension of $d_{\text{model}} = 2304$. **For FFN**, Gemma-2 uses the Gated FFN in Eq. (12) with $d'_{\text{ff}} = 9216$. We replace it with Spark FFN in Eq. (9) with $d_{\text{ff}} = 13824$ so that the parameter count keeps the same. In addition, we take $k = 1106$, which gives a sparsity level of $8\%$, and $r = 1024$ (due to sharding constraints, $r$ can only be a multiple of 256). **For Attention**, Gemma-2 alternates between a global attention that have a span of 8192 tokens, and a local attention with a 4096 window size, both with $d_{\text{attn}} = 256$. We replace both with Spark Attention in Eq. (13) where for the latter we use the same 4096 window size. For hyper-parameters, we use $k = 256$, i.e. each token attends to at most 256 tokens, and $r = 128$. Extra care need to be taken for handling position embedding. Gemma-2 uses Rotary Position Embedding (Su et al., 2024) which is applied to $\boldsymbol{q}$ and the columns of $\boldsymbol{K}$ in Eq. (13). For Spark Attention in Eq. (14), we apply this position encoding to $\boldsymbol{P}\boldsymbol{q}$, $(\boldsymbol{I} - \boldsymbol{P})\boldsymbol{q}$, the columns of $\boldsymbol{P}\boldsymbol{K}$, and the columns of $(\boldsymbol{I} - \boldsymbol{P})\boldsymbol{K}$.

### 4.1 QUALITY

We evaluate Spark Gemma-2 2B on a suite of benchmarks that are used in the Gemma-2 paper (Gemma Team, 2024), and report the result in Table 2. We observe that Spark Gemma 2 matches the quality of Gemma 2 while having a drastically reduced FLOP count per token.

**Sparsity.** To verify the effectiveness of statistical top-$k$, we report the level of sparsity measured in terms of percentage of nonzeros in FFN and the number of nonzeros in Attention. At the beginning of model training, we observe that statistical top-$k$ produces close to $8\%$ nonzeros in FFN (see Figure 1a), which aligns well with our hyper-parameter choice of using $k/d_{\text{ff}} = 8\%$ in Spark FFN. This

Table 2: Evaluation of Spark Transformer quality. We train a *Spark Gemma-2* model, replacing the standard FFN and Attention in Gemma-2 with Spark FFN and Spark Attention, respectively. We compare Spark Gemma-2 with ProSparse and LLaMA ReGLU, two recent models employing activation sparsity in their FFNs. Numbers in parentheses are taken from the respective original papers. FLOPs per token are computed assuming a context length of 8k.

|  | ProSparse (Song et al., 2024a) | LLaMA ReGLU (Zhang et al., 2024) | Gemma-2 | Spark Gemma-2 (Ours) |
|---|---|---|---|---|
| Model size | 7B | 7B | 2B | 2B |
| FLOPs / token | - | - | 4.2B | 1.4B |
| MMLU | (45.5) | (44.8) | 52.1 (52.2) | 50.2 |
| ARC-C | - | - | 50.1 (55.7) | 51.1 |
| GSM8K | (12.1) | (10.6) | 21.2 (24.3) | 21.2 |
| AGIEval | (27.5) | - | 31.8 (31.5) | 31.4 |
| BBH | (35.0) | - | 41.3 (41.9) | 38.8 |
| Winogrande | - | (69.4) | 68.7 (71.3) | 67.3 |
| HellaSwag | - | (74.7) | 73.9 (72.9) | 73.2 |
| MATH | - | - | 16.4 (16.0) | 15.6 |
| ARC-e | - | - | 80.6 (80.6) | 81.3 |
| PIQA | - | - | 78.5 (78.4) | 78.6 |
| SIQA | - | - | 51.6 (51.9) | 51.3 |
| Boolq | - | - | 72.9 (72.7) | 73.3 |
| TriviaQA | - | - | 60.4 (60.4) | 59.4 |
| NQ | - | - | 17.1 (17.1) | 17.1 |
| HumanEval | - | - | 4.3 (20.1) | 6.1 |
| MBPP | - | - | 30.4 (30.2) | 29.0 |
| Avg. | - | - | 47.0 | 46.6 |

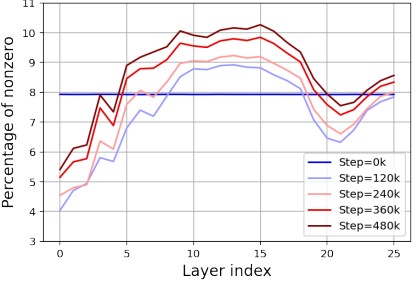

(a) Spark FFN sparsity

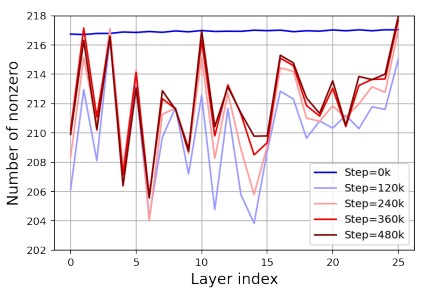

(b) Spark Attention sparsity

Figure 1: Sparsity in the intermediate activation of Spark FFN (i.e., output of GELU) and intermediate activation of Spark Attention (i.e., output of softmax) across all 26 layers at selected training steps. For FFN we report the percentage of nonzero entries out of $d_{\text{ff}} = 13824$ entries. For Attention, we report the number of nonzero entries (i.e., attended tokens). Note that our hyper-parameter choice is to have 8% nonzeros in Spark FFN and at most 256 nonzeros in Spark Attention.

is expected as the model parameters, particularly $K$ in Spark FFN, are randomly initialized, hence the entries of the activation maps are drawn from a Gaussian distribution which is in accordance with the assumption of statistical top-$k$. The Gaussian assumption is no longer guaranteed after training, but we empirically observe it to hold approximately (see Section D.1) and statistical top-$k$ reliably produce a sparsity level close to 8% until the end of training at $480k$ steps. Sparsity in attention is reported in Figure 1b, which show that the number of attended tokens is below our hyper-parameter choice of 256 in Spark Attention throughout training. In particular, the numbers are much smaller because the results are from averaging over all tokens many of which have a context length of less than 256. Finally, we observe comparable levels of sparsity during evaluation (see Section D.2).

## 4.2 EFFICIENCY

We evaluate the efficiency benefits of the Spark Gemma-2 2B over standard Gemma-2 2B using the `gemma.cpp` (Google Gemma.cpp, 2024), a C++ inference engine optimized for the Gemma models

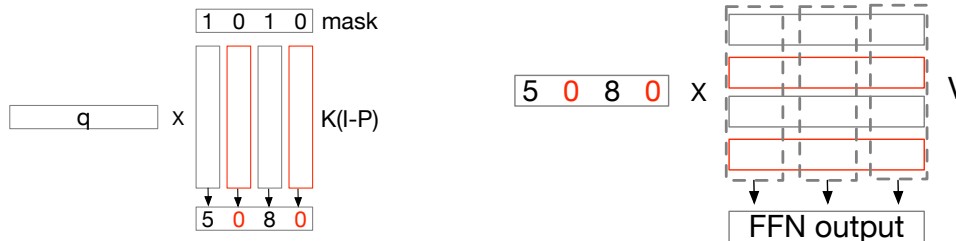

(a) Vector-Masked Matrix Multiplication.

(b) Sparse Vector-Matrix Multiplication.

Figure 2: Illustration of the matrix multiplication implementation using sparse activation. (a) Vector-Masked Matrix Multiplication takes a dense vector $q$, a dense matrix $K^\top(I - P)$, and a mask from statistical top-$k$ on $K^\top Pq$ to compute $u := (K^\top(I - P)q) \odot$ mask. It skips memory loading and compute associated with the masked columns. (b) Sparse Vector-Matrix Multiplication takes a sparse activation vector $u$ to compute weighted sum of rows in the dense matrix $V$. It skips loading and computation of rows corresponding to 0's in $u$. To optimize performance, we implement Sparse Vector-Matrix Multiplication using tiling, which helps minimize cross-CPU core synchronization.

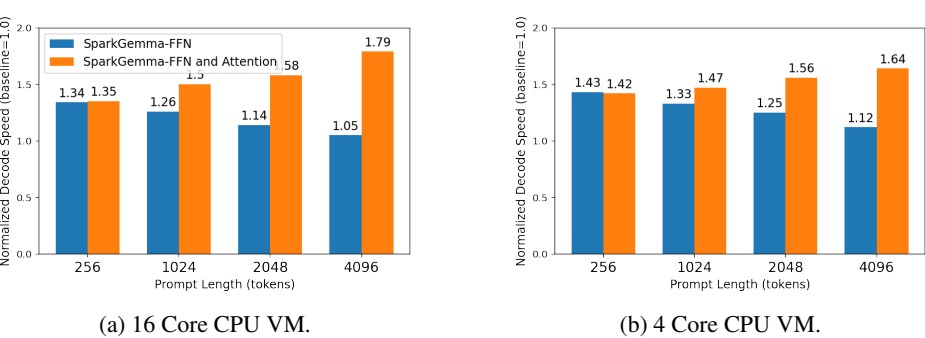

(a) 16 Core CPU VM.

(b) 4 Core CPU VM.

Figure 3: Spark Gemma-2 2B decoding speedup on CPU relative to the original Gemma-2 2B for varying prompt lengths. Speed is measured as the decoding time per token average over 128 tokens after the prompt. We provide a breakdown of speedup from FFN and Attention by reporting results of SparkGemma-FFN, which contains sparse optimization for FFN only, and SparkGemma-FFN and Attention, which contains sparse optimization for both. We use decode batch size of one.

on CPUs. Our implementation uses sparse matrix multiplication operators, which exploit sparsity in both FFN and Attention, as well as modern CPU vector SIMD operations (SIMD Wikipedia, 2024), see Figure 2 for an illustration and Section C in Appendix for details. We show that Spark Gemma-2 significantly improves the efficiency of transformer models, even in highly FLOP-constrained environments such as CPUs.

Specifically, Figure 3 reports the decoding speed under varying prompt lengths on a 4-Core or a 16-Core CPU. We see that Spark Gemma-2 outperforms the original Gemma-2 model, achieving a speedup that ranges from 1.35x to 1.79x on 16-Core CPU depending on the prompt length. For short prompts (e.g., 256 tokens), the sparse FFN optimization provides most of the speedup, whereas the sparse attention optimization provides the most speedup for longer prompts (e.g. 4096 tokens).

Table 3 further highlights the efficiency of Spark Gemma-2 in both prefill and decode phases. During the prefill, the prompt is usually chunked into batches since the process is bounded by memory bandwidth. This may reduce the benefit of activation sparsity as different tokens in a chunk may activate different subsets of parameters (in FFN) and attend to different subsets of tokens (in Attention). However, Table 3 shows that Spark Gemma-2 maintains strong performance with a chunk size of 64 tokens, following the default setup in gemma.cpp. A more detailed performance analysis of batching/chunking is provided in the Appendix. In addition, Spark Gemma-2 significantly outperforms the original Gemma-2 during decoding (with batch size=1). Notably, it achieves a decode speed of 86ms per token, which surpasses the average human reading speed (238 words per minute) (Brysbaert, 2019), with a very accessible 4-Core CPU Cloud VM.

Table 3: Prefill and decode speed of Spark Gemma-2 on 4-Core and 16-Core CPUs for prompts of 4096 tokens. During prefill phase, the prompt is chunked into batches of 64 tokens, following a default setup of `gemma.cpp`. Speedups relative to Gemma-2 are shown in parentheses.

|  | Prefill (ms / token) | Decode (ms / token) |
| --- | --- | --- |
| 4 Core CPU VM | 15 (1.86x) | 86 (1.64x) |
| 16 Core CPU VM | 7 (1.7x) | 33 (1.79x) |

### 4.3 ABLATION

**Statistical top-$k$.** Adding a top-$k$ operator is expected to lead to a training slowdown due to the extra computation cost of computing top-$k$. In Figure 4 we report the slowdown during training due to adding statistical top-$k$. It can be observed that the slowdown is with a very small amount, demonstrating the efficiency of statistical top-$k$. In particular, we compare the slowdown with that of the top-$k$ operator provided in `JAX`, namely `jax.lax.approx_max_k` (Chern et al., 2022). This operator is optimized to achieve TPU peak performance and has a controllable recall target, which we vary on the x-axis. Statistical top-$k$ is significantly faster than the `JAX` top-$k$ even when the latter operates on a small recall of 50%. Finally, we do not provide the quality of models trained with `JAX` top-$k$ since such models take a very long time to train.

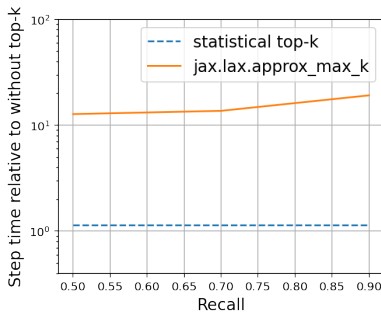

Figure 4: Comparison of training slowdown from using our statistical top-$k$ vs the standard top-$k$ (i.e., `jax.lax.approx_max_k` (Chern et al., 2022)) relative to not using any top-$k$.

**Effect of $r$ and $k$ in Spark FFN.** Spark FFN comes with two hyper-parameters, namely $r$ which controls the rank hence FLOP count of the low-cost predictor, and $k$ which controls sparsity of activation hence the FLOP count. In Figure 5 we provide an ablation study on the effect of these two hyper-parameters, by reporting the training loss curves in the first 25,000 training steps (which is around 5% of full training). From Figure 5a, the best choice of $r$ is 1024 which is nearly half of $d_{\text{model}} = 2304$ (due to model sharding constraint, $r$ cannot be taken to be exactly a half of $d_{\text{model}}$). From Figure 5b, we see that the model quality is insensitive to choices of $k$ that gives $[5\%, 10\%]$ sparsity, but there is quality loss if we go sparser, e.g. 3% nonzeros.

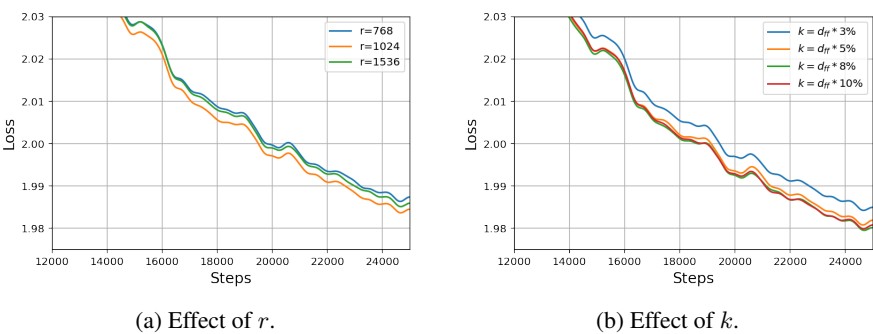

(a) Effect of $r$.

(b) Effect of $k$.

Figure 5: Effect of hyper-parameters $r$ and $k$ in Spark FFN on training loss. A Gaussian filter of $\sigma = 200$ is applied to smooth the loss curves.

## 5 DISCUSSION AND RELATED WORK

Returning to the question posed at the outset: How many FLOPs is a token worth? This paper offers an answer through the Spark Transformer architecture, demonstrating at least a $3\times$ overall FLOPs reduction without sacrificing model quality. This reduction is realized by selectively activating only part of the model parameters and limiting the attended context for each input. This principle of sparse activation finds a compelling parallel in neuroscience, where studies reveal sparse activity patterns in the brain as a key factor in its remarkable efficiency (Attwell & Laughlin, 2001; Barth & Poulet, 2012; Lee et al., 2024a). While hardware limitations currently hinder the full exploitation of

sparse activation in Transformers, particularly on GPUs and TPUs designed for dense computations, our work with Spark Transformer on CPUs highlights its potential. We believe this opens avenues for research into alternative hardware and platforms better suited for sparse computations, hence circumvents the hardware lottery (Hooker, 2021) and potentially lead to greater efficiency gains in the future.

In the following, we review a few lines of work closely related to ours.

**Mixture of Experts (MoEs)** may be considered as a particular case of sparsely activated models which group the neurons in FFN and activate all neutrons in selected groups (Shazeer et al., 2017; Lepikhin et al., 2020). Neuron grouping has the benefit of being better suited for training accelerators compared to unstructured activation sparsity. However, training of MoEs incurs extra complexities in algorithmic design and requires special hardware support (Fedus et al., 2022). Moreover, the structured nature of sparsity limits the model's flexibility and expressiveness, and recent work advocates the use of a larger number of smaller experts (Dai et al., 2024; He, 2024). On the other hand, the discovery of the naturally emerging unstructured activation sparsity has motivated the new perspective of naturally emerging experts (Zhang et al., 2022; Dong et al., 2023; Csordás et al., 2023; Qiu et al., 2023; Szatkowski et al., 2024; Zheng et al., 2024).

**Sparse activation** is common approach to improve the efficiency of large models and many techniques for a low-cost activation prediction have been developed over the years, such as low-rank factorization (Davis & Arel, 2013), quantization (Cao et al., 2019), product keys (Lample et al., 2019), hashing (Chen et al., 2020), etc. With the popularity of modern large Transformer models, these techniques become natural choices (Jaszczur et al., 2021; Zeng et al., 2023; Liu et al., 2023; Song et al., 2023) for reducing their high computation costs. In particular, a lot of the excitement comes from the discovery that the activations in FFNs are naturally sparse (Zhang et al., 2022; Li et al., 2022) and hence efficiency with activation sparsity are obtained without a quality toll.

Our work falls into the category of the latest work in this direction that aim to bring the benefits to the latest generation large language models that do not have natural sparsity. Early attempts (Mirzadeh et al., 2023; Peng et al., 2023; Zhang et al., 2024) seek to bring back sparsity by switching back to ReLU variants, but it usually incurs a quality loss. The quality gap may be largely bridged by more careful tuning, but the activation becomes less sparse (e.g. $25\%$ nonzeros in LLAMA 7B (Song et al., 2024a)). Top-k becomes a more popular choice for obtaining sparsity recently (Song et al., 2024b) and is able to maintain neutral quality while offering strong sparsity, but only in selected layers (Yerram et al., 2024). Moreover, such methods require finetuning to bring sparsity and also obtain a predictor. Without doing finetuning, Lee et al. (2024b); Liu et al. (2024a) obtained at most 50% nonzeros under neutral quality. In contrast to these works, our work not only obtains 8% nonzeros in activation of all FFN layers, but also a predictor, all with a single-stage training. We provide a summary of comparison to these methods in Table 4 in the Appendix.

Finally, the usefulness of activation sparsity goes beyond efficiency. For example, theoretical studies show its benefits for model generalizability and learnability (Muthukumar & Sulam, 2023; Awasthi et al., 2024). Moreover, activated neurons may be associated with semantic concepts, which offers understanding of the working mechanism and enables manipulating the output of Transformer models (Cuadros et al., 2022; Luo et al., 2024).

**Sparse attention** broadly refers to the approach of attending to a selected subset of tokens in the context as a means of reducing computation cost (Deng et al., 2024; Jiang et al., 2024). Work on sparse attention include those that use handcrafted attention patterns (Child et al., 2019; Beltagy et al., 2020; Ainslie et al., 2023; Ding et al., 2023), which feature simplicity, and learned attention patterns (Kitaev et al., 2020; Roy et al., 2021) which feature better modeling capacity. However, learning attention patterns often involve learning, e.g., a hash table or k-means centers, which significantly complicates modeling. Closely related to our Spark Attention is the top-$k$ attention (Gupta et al., 2021), which obtains data-adaptive attention simply from top-$k$ thresholding. Our work improves upon top-$k$ attention by introducing a low cost predictor which enables an increased computational benefits from sparsity. Finally, KV pruning approaches drop selected tokens permanently as decoding proceeds (Zhang et al., 2023; Liu et al., 2024b), and cannot achieve as high compression ratio as Top-$k$ based approaches.

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

# A    PROOF TO THEOREM 1

*Proof.*  In this proof we write $\bar{x} \stackrel{\text{def}}{=} \text{mean}(x)$ and $s \stackrel{\text{def}}{=} \text{std}(x)$ for brevity.

We first establish the concentration bounds that the empirical mean and standard deviation, i.e., $\bar{x}$ and $s$ are close to the true mean and true standard deviation, i.e., $\mu$ and $\sigma$ of the underlying Gaussian, respectively. Recall from the definition of the chi-squared distribution that $(d-1)\frac{s^2}{\sigma^2} \sim \chi^2(d-1)$. Using the Laurent-Massart bound on the tail probability of the chi-squared distribution (Laurent & Massart, 2000, Corollary of Lemma 1), we have

$$\Pr\left(\left|(d-1)\frac{s^2}{\sigma^2} - (d-1)\right| \geq 2\sqrt{(d-1)t} + 2t\right) \leq 2e^{-t}$$

for every $t > 0$. We set $t = \log\frac{6}{\delta}$. Then, with a probability of at least $1 - \delta/3$, we have

$$(d-1)\left|\frac{s^2}{\sigma^2} - 1\right| < 2\sqrt{(d-1)\log\frac{6}{\delta}} + 2\log\frac{6}{\delta},$$

which implies

$$\left|\frac{s^2}{\sigma^2} - 1\right| < 2\sqrt{\frac{\log\frac{6}{\delta}}{d-1}} + 2\frac{\log\frac{6}{\delta}}{d-1} \leq 4\sqrt{\frac{\log\frac{6}{\delta}}{d}} + 4\frac{\log\frac{6}{\delta}}{d} \leq 8\sqrt{\frac{\log\frac{6}{\delta}}{d}},$$

where the last inequality uses the assumption that $d \geq \max\{2, \log\frac{6}{\delta}\}$. By rearranging the terms, we get

$$\sigma\left(1 - 8\sqrt{\frac{\log\frac{6}{\delta}}{d}}\right) \leq \sigma\sqrt{\max\left\{1 - 8\sqrt{\frac{\log\frac{6}{\delta}}{d}}, 0\right\}} \leq s \leq \sigma\sqrt{1 + 8\sqrt{\frac{\log\frac{6}{\delta}}{d}}} \leq \sigma\left(1 + 8\sqrt{\frac{\log\frac{6}{\delta}}{d}}\right),$$

which simplifies to

$$|s - \sigma| \leq 8\sigma\sqrt{\frac{\log\frac{6}{\delta}}{d}}. \tag{17}$$

Eq. (17) provides a concentration bound for $s$. We now proceed to deriving a bound for $\mu$. Towards that, notice that $\frac{\bar{x}-\mu}{\sigma/\sqrt{d}} \sim \mathcal{N}(0,1)$. By using the Mill's inequality that upper bounds the tail probability of a standard normal distribution (i.e., if $Z \sim \mathcal{N}(0,1)$ and $t > 0$, then $\Pr(|Z| > t) \leq \frac{e^{-t^2/2}}{t}$), we have

$$\Pr\left(\left|\frac{\bar{x}-\mu}{\sigma/\sqrt{d}}\right| > \sqrt{2\log\frac{3}{\delta}}\right) \leq \frac{\delta/3}{\sqrt{2\log\frac{3}{\delta}}} \leq \delta/3.$$

Therefore, with probability at least $1 - \delta/3$, we have

$$\left|\frac{\bar{x}-\mu}{\sigma/\sqrt{d}}\right| \leq \sqrt{2\log\frac{3}{\delta}},$$

which yields

$$|\bar{x} - \mu| \leq \sigma\sqrt{\frac{2\log\frac{3}{\delta}}{d}}. \tag{18}$$

Combining Eq. (17) and Eq. (18), with probability at least $1 - 2\delta/3$, we have

$$\left|\theta(\boldsymbol{x}, k) - (\mu + \sigma Q(1 - \frac{k}{d}))\right| \tag{19}$$

$$\leq |\bar{x} - \mu| + |s - \sigma|\left|Q(1 - \frac{k}{d})\right| \tag{20}$$

$$\leq \sigma\sqrt{\frac{2\log\frac{3}{\delta}}{d}} + 8\sigma\sqrt{\frac{\log\frac{6}{\delta}}{d}}\left|Q(1 - \frac{k}{d})\right|. \tag{21}$$

We define the empirical cumulative distribution function (ECDF) of $x_1, x_2, \ldots, x_d$ as

$$\hat{F}_d(x) = \frac{1}{d} \sum_{i \in [d]} \mathbf{1}_{\{x_i \leq x\}}.$$

Then, the number of the entries of $\boldsymbol{x}$ that are greater than $\theta(\boldsymbol{x}, k)$ may be written as

$$\mathbf{card}\left(\{i \in [d] \mid x_i > \theta(\boldsymbol{x}, k)\}\right) = \sum_{i \in [d]} \mathbf{1}_{\{x_i > \theta(\boldsymbol{x}, k)\}} = d\left(1 - \hat{F}_d(\theta(\boldsymbol{x}, k))\right).$$

Let $F$ denote the cumulative distribution function (CDF) of $\mathcal{N}(\mu, \sigma^2)$. By the Dvoretzky-Kiefer-Wolfowitz inequality (Dvoretzky et al., 1956; Massart, 1990), we have

$$\Pr\left(\sup_{u \in \mathbb{R}} \left|\hat{F}_d(u) - F(u)\right| > t\right) \leq 2e^{-2dt^2}.$$

Taking $t = \sqrt{\frac{1}{2d} \log \frac{6}{\delta}}$ and $u = \theta(\boldsymbol{x}, k)$, we obtain

$$\Pr\left(\left|\hat{F}_d(\theta(\boldsymbol{x}, k)) - F(\theta(\boldsymbol{x}, k))\right| > \sqrt{\frac{1}{2d} \log \frac{6}{\delta}}\right) \leq \frac{\delta}{3}. \tag{22}$$

Applying the union bound on Eq. (19) and Eq. (22), we obtain that the following holds with probability at least $1 - \delta$:

$$\left|\hat{F}_d(\theta(\boldsymbol{x}, k)) - (1 - \frac{k}{d})\right| \tag{23}$$

$$= \left|\hat{F}_d(\theta(\boldsymbol{x}, k)) - F(\mu + \sigma Q(1 - \frac{k}{d}))\right| \tag{24}$$

$$\leq \left|\hat{F}_d(\theta(\boldsymbol{x}, k)) - F(\theta(\boldsymbol{x}, k))\right| + \left|F(\theta(\boldsymbol{x}, k)) - F(\mu + \sigma Q(1 - \frac{k}{d}))\right| \tag{25}$$

$$\leq \sqrt{\frac{1}{2d} \log \frac{6}{\delta}} + \frac{1}{\sqrt{2\pi}\sigma} \left|\theta(\boldsymbol{x}, k) - (\mu + \sigma Q(1 - \frac{k}{d}))\right| \tag{26}$$

$$\leq \sqrt{\frac{1}{2d} \log \frac{6}{\delta}} + \frac{1}{\sqrt{2\pi}} \left(\sqrt{\frac{2 \log \frac{3}{\delta}}{d}} + 8\sqrt{\frac{\log \frac{6}{\delta}}{d}} \left|Q(1 - \frac{k}{d})\right|\right) \tag{27}$$

$$\leq 4\sqrt{\frac{\log \frac{6}{\delta}}{d}} \left(1 + \left|Q(1 - \frac{k}{d})\right|\right). \tag{28}$$

In the above expression, the first equality stems directly from the definitions of $F(\cdot)$ and $Q(\cdot)$, which gives

$$F(\theta(\boldsymbol{x}, k)) = F(\mu + \sigma Q(1 - \frac{k}{d})) = \Phi(Q(1 - \frac{k}{d})) = 1 - \frac{k}{d},$$

where $\Phi$ denotes the CDF of the standard normal distribution.

To simplify Eq. (23), we consider two cases:

- If $k \leq d/2$, by Mill's inequality, we have

$$1 - \Phi(\sqrt{2 \log \frac{d}{k}}) \leq \frac{e^{-(\sqrt{2 \log \frac{d}{k}})^2/2}}{\sqrt{2 \log \frac{d}{k}}} = \frac{e^{-(\sqrt{2 \log \frac{d}{k}})^2/2}}{\sqrt{2 \log \frac{d}{k}}} = \frac{k/d}{\sqrt{2 \log \frac{d}{k}}} \leq \frac{k}{d},$$

  where the last inequality is because $2 \log \frac{d}{k} \geq 1$. Therefore

$$1 - \frac{k}{d} \leq \Phi(\sqrt{2 \log \frac{d}{k}}),$$

  which gives

$$Q(1 - \frac{k}{d}) \leq \sqrt{2 \log \frac{d}{k}} = \sqrt{-2 \log \frac{k}{d}}.$$

- If $k > d/2$, we have

$$\left| Q(1 - \frac{k}{d}) \right| = \left| Q(1 - \frac{d-k}{d}) \right| \leq \sqrt{-2 \log \frac{d-k}{d}} \, .$$

Combining the two cases, we get

$$\left| Q(1 - \frac{k}{d}) \right| \leq \sqrt{-2 \log \min \left\{ \frac{k}{d}, 1 - \frac{k}{d} \right\}} \, .$$

Plugging this into Eq. (23), we obtain

$$\left| \hat{F}_d(\theta(\boldsymbol{x}, k)) - (1 - \frac{k}{d}) \right| \leq 4 \sqrt{\frac{\log \frac{6}{\delta}}{d}} \left( 1 + \sqrt{-2 \log \min \left\{ \frac{k}{d}, 1 - \frac{k}{d} \right\}} \right) .$$

Recall $\mathbf{card} \left( \{ i \in [d] \mid x_i > \theta(\boldsymbol{x}, k) \} \right) = d \left( 1 - \hat{F}_d(\theta(\boldsymbol{x}, k)) \right)$. We conclude that with probability at least $1 - \delta$, we have

$$\left| \mathbf{card} \left( \{ i \in [d] \mid x_i > \theta(\boldsymbol{x}, k) \} \right) - k \right| \leq 4 \sqrt{d \log \frac{6}{\delta}} \left( 1 + \sqrt{-2 \log \min \left\{ \frac{k}{d}, 1 - \frac{k}{d} \right\}} \right) .$$

$\square$

## B    PROOF TO THEOREM 2

*Proof.* The Huber statistical top-$k$ in Eq. (4) may be written as

$$\text{Huber}(\text{Statistical-Top}_k(\boldsymbol{x}); \delta)/\delta = \text{Huber}(\text{Soft-Threshold}(\boldsymbol{x}, \theta(\boldsymbol{x}, k)))/\delta, \tag{29}$$

where $\theta(\boldsymbol{x}, k)$ is defined in Eq. (2). This function is the (multivariate) composition of two functions, namely, $\theta = \theta(\boldsymbol{x}, k)$ and $\text{Huber}(\text{Soft-Threshold}(\boldsymbol{x}, \theta))$. In particular, the former is continuously differentiable (i.e., $C^1$) in $\boldsymbol{x}$, since it is simply a linear combination of sample mean and sample standard deviation both of which are $C^1$ functions. To establish the theorem, we only need to show that $\text{Huber}(\text{Soft-Threshold}(\boldsymbol{x}, \theta))$ is also a $C^1$ function in $(\boldsymbol{x}, \theta)$.

By definition, $\text{Huber}(\text{Soft-Threshold}(\boldsymbol{x}, \theta))$ is defined entry-wise on $\boldsymbol{x}$ as

$$\text{Huber}(\text{Soft-Threshold}(x, \theta)) = \begin{cases} \delta x - \delta \theta - \frac{1}{2} \delta, & \text{if } x > \theta + \delta; \\ \frac{1}{2}(x - \theta)^2, & \text{if } \theta \leq x \leq \theta + \delta; \\ 0, & \text{if } x < \theta. \end{cases} \tag{30}$$

From here it is easy to check that $\text{Huber}(\text{Soft-Threshold}(x, \theta))$ is continuous in $(x, \theta)$. Its gradient with respect to $(x, \theta)$ is given by

$$\frac{\partial \text{Huber}(\text{Soft-Threshold}(x, \theta))}{\partial(x, \theta)} = \begin{cases} (\delta, -\delta), & \text{if } x > \theta + \delta; \\ (x - \theta, \theta - x) & \text{if } \theta \leq x \leq \theta + \delta; \\ (0, 0), & \text{if } x < \theta, \end{cases} \tag{31}$$

which is also continuous. This concludes the proof. $\square$

## C    IMPLEMENTATION DETAILS ON SPARSE MATRIX MULTIPLICATIONS

We describe how we implement sparse matrix multiplications for Spark FFN and Attention in `gemma.cpp`. We start by focusing on a batch size of one for decoding before expanding our discussion to larger batch sizes and prefill.

With batch size of 1, both Spark FFN and Spark Attention utilize two types of sparse vector-matrix multiplication: vector-masked matrix multiplication and sparse vector-matrix multiplication (Figure 2). Given a vector $q$ and a matrix $w$, vector-masked matrix multiplication multiplies $q$ with the non-masked columns of $w$ based on a masking vector $m$. Masked columns yield a zero output. Sparse vector-matrix multiplication, on the other hand, involves a vector that contains many zeroes being multiplied by a dense matrix.

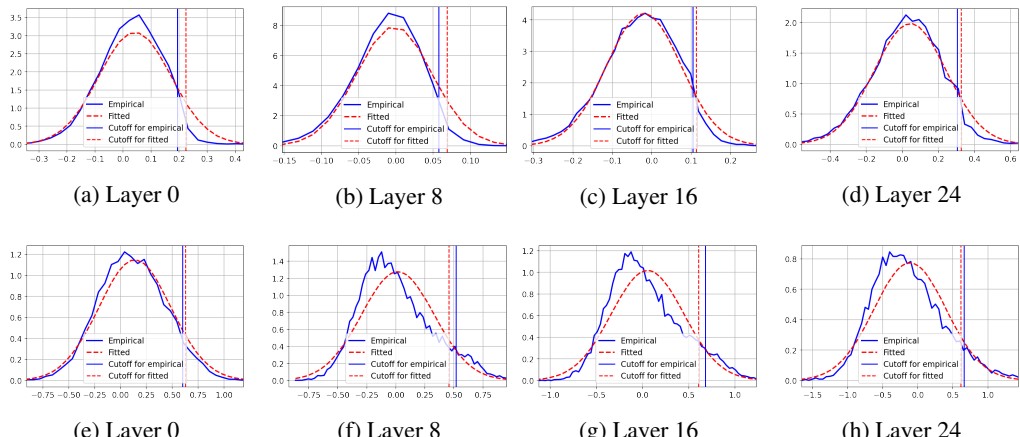

(a) Layer 0    (b) Layer 8    (c) Layer 16    (d) Layer 24

(e) Layer 0    (f) Layer 8    (g) Layer 16    (h) Layer 24

Figure 6: Distribution of the entries of the input activation to statistical top-$k$ in Spark FFN (see Figure 7 for result of Spark Attention). The two rows correspond to activation at two positions 0 and 1000 of an input, and the columns correspond to activation at four different depth levels $\{0, 8, 16, 24\}$ of the 26-layer pretrained Spark Gemma-2. The input is the first 1000 tokens of the first essay from `https://paulgraham.com/articles.html` prepended with the BOS token. We compare the empirical distribution (*Empirical*) with the Gaussian distribution whose mean and standard deviation (std) are computed as the sample mean and std of the input (*Fitted*). *We see that the Gaussian closely approximates the empirical distribution.* We also compare the cutoff value estimated from the Gaussian, i.e., $\theta(\boldsymbol{x}, k)$ used in Eq. (2) with $k/d = 5\%$ (*Cutoff for fitted*), with the cutoff value for obtaining $8\%$ nonzeros on the empirical distribution (*Cutoff for empirical*). *It can be seen that these two cutoff values are close.*

In Spark FFN, we perform vector-masked matrix multiplication for $\boldsymbol{K}^\top(\boldsymbol{I} - \boldsymbol{P})\boldsymbol{q}$ (Figure 2a). The masking vector is generated from the output of $\text{Statistical-Top}_k(\boldsymbol{K}^\top \boldsymbol{P}\boldsymbol{q})$. In the CPU implementation, the columns of $w$ are loaded from DRAM one at a time. Based on the mask, Spark FFN skips loading the masked columns of $w$ from DRAM and the associated computations. Spark FFN utilizes SIMD operations (as in the original Gemma implementation). To further enhance perforamnce, Spark FFN utilizes software CPU prefetching ($builtin\_prefetch$) to overlap loading from DRAM to the CPU cache with computations.

The same masking vector also identifies the zero entries in the intermediate vector that is multiplied by matrix $V$ (Figure 2b). For this sparse vector-matrix multiplication, we store the matrix in row format. Each CPU thread processes a tile of the matrix while skipping the loading and computation of the masked rows. Prefetching and SIMD operations are applied similarly in this context.

Spark Attention utilizes these two types of sparse matrix multiplication operators to accelerate qkv computations for each head.

When extending to decoding with batch sizes greater than one or prefill, we continue to use individual masks to skip *computations* while using a union of masks from each vector within the batch to create unified masks for *memory loading*. With larger batches, Spark transformer is expected to save less memory loading (vs. original Gemma), unless there is significant overlap in top-k positions within the same batch. Nonetheless, the Spark transformer consistently reduces FLOP by skipping computations based on individual masks within the batch.

# D  ADDITIONAL EXPERIMENTAL RESULTS

## D.1  DISTRIBUTION OF INPUTS TO STATISTICAL TOP-$k$

The underlying assumption for statistical top-$k$ is that the activation vector upon which it is applied to, namely, the pre-GELU activation in Spark FFN and the pre-softmax activation in Spark Attention, can be modeled as being drawn from an i.i.d. Gaussian distribution. Here we provide empirical evaluation on the distribution of these activation vectors for Spark Gemma2. Results for Spark FFN

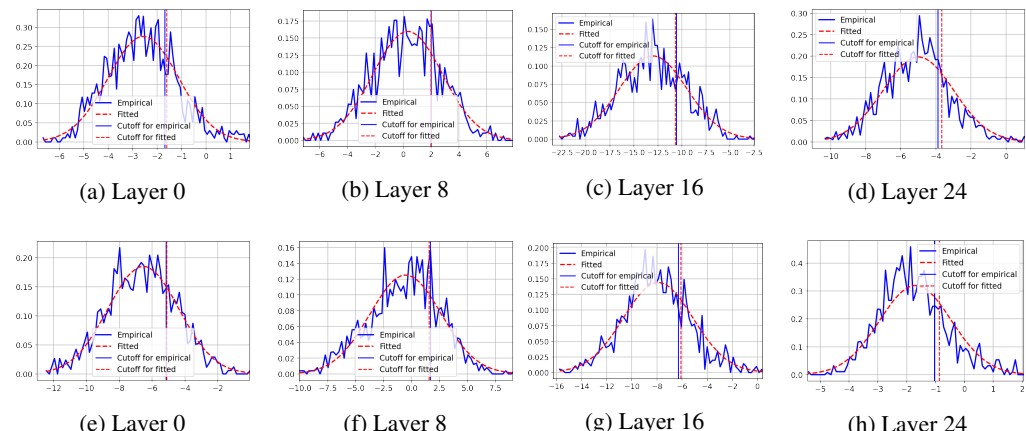

(a) Layer 0 (b) Layer 8 (c) Layer 16 (d) Layer 24

(e) Layer 0 (f) Layer 8 (g) Layer 16 (h) Layer 24

Figure 7: Distribution of the entries of the input activation to statistical top-$k$ in Spark Attention (see Figure 6 for result of Spark FFN). The two rows correspond to activation for two different attention heads, and the columns correspond to activation at four different depth levels $\{0, 8, 16, 24\}$ of the 26-layer pretrained Spark Gemma-2. Model input is the first 1000 tokens of the first essay from `https://paulgraham.com/articles.html` prepended with the BOS token, and we examine activation of the last token (i.e., inner product between the query embedding of the 1001st token and all 1001 key embeddings). We compare the empirical distribution (*Empirical*) with the Gaussian distribution whose mean and standard deviation (std) are computed as the sample mean and std of the input (*Fitted*). *We see that the Gaussian closely approximates the empirical distribution.* We also compare the cutoff value estimated from the Gaussian, i.e., $\theta(\boldsymbol{x}, k)$ used in Eq. (2) with $k = 256$ (*Cutoff for fitted*), with the cutoff value for obtaining top 256 entries on the empirical distribution (*Cutoff for empirical*). *It can be seen that these two cutoff values are close.*

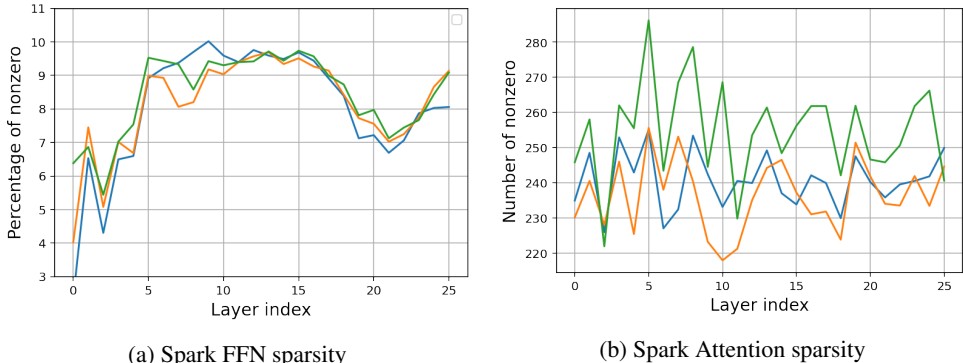

(a) Spark FFN sparsity (b) Spark Attention sparsity

Figure 8: Sparsity in the intermediate activation of Spark FFN and Spark Attention *during evaluation* (see Figure 1 for results during training). For FFN, we use a simple prompt "test" and report the percentage of nonzero entries in generating the 5th, 10th, and 15th token. For Attention, we report the nuber of nonzero entries at the 512th, 1024th, and 2048th token during prefill.

and Spark Attention are provided in Figure 6 and Figure 7, respectively. The results show that the distribution holds close proximity to a Gaussian, hence justifying the use of statistical top-$k$.

## D.2 SPARSITY LEVEL DURING EVALUATION

Complementing Figure 1 which reports sparsity level during pretraining, here we report the sparsity level during evaluation to confirm that statistical top-$k$ produces the same level of sparsity during test time. The results are presented in Figure 8 for some arbitrarily selected tokens. For Attention, in particular, we select tokens at the positions 512, 1024, and 2048 which are all above our choice of $k = 256$ for Spark Attention.

## D.3 BATCHING ANALYSIS

Figure 9 provides the performance comparison between Spark Gemma 2 and Gemma 2, measured in prefill throughput (tokens/sec) across varying chunk sizes. We use a 4096-token prompt on a 16-core CPU VM. A similar trend is expected during the decoding phase with varying batch sizes.

Our analysis shows that Spark Gemma-2 provides the highest speedup at batch size 1, and again at large batch sizes (e.g. >8), where the compute FLOP becomes the primary bottleneck.

For Gemma-2, as seen in the figure, increasing batch/chunk size leads to a significant

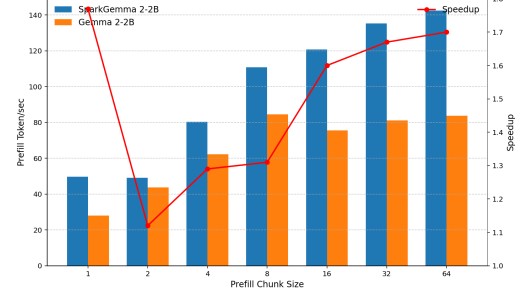

Figure 9: Spark Gemma-2 vs. Gemma-2 Prefill Token/Sec with Varying Chunk Sizes. We use a prompt length of 4096 tokens on a 16 core CPU VM.

improvement in prefill throughput until the batch size reaches 8. This improvement occurs because batching reduces memory access by reusing weights across multiple tokens in the CPU cache. Once the computation becomes the bottleneck (i.e. batch = 8), further batching provides diminishing returns.

In contrast, Spark Gemma-2 behaves differently. When the batch size increases from 1 to 2, we observe minimal throughput change. This is due to the lack of overlap in top-k positions between the tokens, resulting from the high sparsity. However, as the batch size increases beyond 4, Spark Gemma-2 starts benefiting from weight reuse, similar to Gemma-2. Spark Gemma-2 continues to show improvements in throughput until the batch size reaches approximately 64, where it eventually becomes FLOP-bound, much later than Gemma-2 due to the reduced FLOP requirements.

Overall, Spark Gemma demonstrates the most significant gains in two scenarios: when the batch size is 1, a common setting for desktop or mobile devices decoding, and when the batch size is large enough that FLOP becomes the dominant bottleneck.

## D.4 ADDITIONAL ABLATION STUDIES

In this section, we provide ablation studies for understanding the effect of the individual components of Spark Transformer. Towards that, we plot the training loss curves for Gemma-2 and Spark Gemma-2, see Figure 10. Here, we restrict to the first 80k training steps out of the 500k total steps since it is costly to fully train all ablation models, and that 80k steps is sufficient for seeing the trend. We can see that Spark Gemma-2 slightly lags behind Gemma-2. However, as demonstrated in Table 2, that small difference in training loss does not lead to a substantial difference in evaluation quality.

In our ablation studies below, we add a single component at a time to Gemma-2 and see the quality impact.

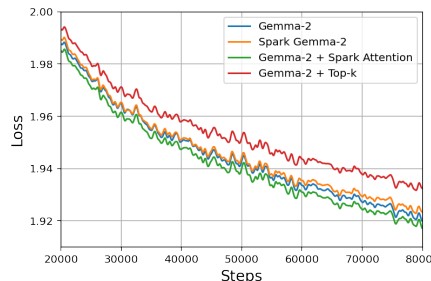

Figure 10: Ablation study in terms of training loss in the first 80k training steps (out of 500k total steps).

**Spark FFN vs Spark Attention.** To understand the effect of Spark FFN vs Spark Attention, we conduct an experiment where only attention is switched from a standard one to Spark Attention, whereas the FFN remains the standard one. The result is illustrated as *Gemma-2 + Spark Attention* in Figure 10. It can be seen that Spark Attention provides a minor quality gain over Gemma-2. In comparing *Gemma-2 + Spark Attention* with Spark Gemma-2, this also shows that further adding Spark FFN slightly hurts model quality. As noted above, such a small difference does not lead to substantial quality impact on the evaluation tasks. Hence, we conclude here that none of Spark FFN and Spark Attention has significant quality impact.

**Sparsity enforcing vs Low-cost predictor.** Sparsity enforcing via statistical top-$k$ and low-cost activation predictor are two relatively independent components of Spark Transformer. This means that, upon the standard Gated FFN (see Eq. (12)) that is used in Gemma-2, which we rewrite here for convenience:

$$\text{Gated-FFN}(\boldsymbol{q}; \boldsymbol{K}_1, \boldsymbol{K}_2, \boldsymbol{V}) = \boldsymbol{V} \cdot \left( \sigma\left( \boldsymbol{K}_1^\top \boldsymbol{q} \right) \odot \left( \boldsymbol{K}_2^\top \boldsymbol{q} \right) \right), \tag{32}$$

we may choose to only apply statistical top-$k$ for enforcing sparsity, i.e.,

$$\text{Topk-Gated-FFN}(\boldsymbol{q}; \boldsymbol{K}_1, \boldsymbol{K}_2, \boldsymbol{V}) = \boldsymbol{V} \cdot \left( \sigma\left( \text{Statistical-Top}_k(\boldsymbol{K}_1^\top \boldsymbol{q}) \right) \odot \left( \boldsymbol{K}_2^\top \boldsymbol{q} \right) \right). \tag{33}$$

Note that applying a sparsifying function on the input to the nonlinear function $\sigma()$ as in Eq. (33) is a common choice in the literature of sparse activations, e.g., Mirzadeh et al. (2023); Song et al. (2024a); Lee et al. (2024b); the main difference between these works lies in the specific sparsity enforcing technique, see Table 4 for a summary. In addition to the sparsifying function, Spark FFN also has another architectural change for the purpose of introducing a low-cost predictor. Here, we rewrite Spark FFN for ease of comparison with Eq. (33):

$$\text{Spark-FFN}(\boldsymbol{q}; \boldsymbol{K}, \boldsymbol{V}, k, r) \stackrel{\text{def}}{=} \boldsymbol{V} \cdot \left( \sigma\left( \text{Statistical-Top}_k(\boldsymbol{K}^\top \boldsymbol{P} \boldsymbol{q}) \right) \odot \left( \boldsymbol{K}^\top (\boldsymbol{I} - \boldsymbol{P}) \boldsymbol{q} \right) \right). \tag{34}$$

Analogous to FFN, we may also only add statistical top-$k$ to attention without the low-cost predictor, i.e., by switching from standard Attention in Eq. (13) to the following:

$$\text{Topk-Attention}(\boldsymbol{q}; \boldsymbol{K}, \boldsymbol{V}) \stackrel{\text{def}}{=} \boldsymbol{V} \cdot \text{softmax}\left( \text{Statistical-Top}_k(\boldsymbol{K}^\top \boldsymbol{q}) \right). \tag{35}$$

Here, we aim to understand the effect of introducing statistical top-$k$ without the low-cost predictor. Towards that, we conduct an experiment where FFN and Attention in Gemma-2 are replaced with Eq. (33) and Eq. (35), respectively. The result is illustrated as *Gemma-2 + Top-k* in Figure 10. It can be seen that the training loss becomes notably larger and the gap compared to Gemma-2 is further increasing with more training steps. This result demonstrates that while the low-cost predictor is introduced to Spark Transformer for reducing the cost in predicting the nonzero entries, it also helps in bridging the gap from the introduction of statistical top-$k$. In other words, Transformer with low-rank predictors in FFN and Attention is more amenable to activation sparsification without quality loss.

# E COMPARISON WITH RELATED WORK ON ACTIVATION SPARSITY IN FFN

In Table 4, we provide a summary of recent work on enabling sparse activation in the latest LLMs.

We can see that our Spark Transformer leads to a FLOPs reduction of -72%, which is more than all the other methods. This comes at a cost of -0.9% quality loss, which is lower than most of the other methods (i.e., ReLUification and ProSparse) and is on par with the best alternative, i.e., HiRE.

# F ADDITIONAL DISCUSSION ON STATISTICAL TOP-$k$

## F.1 NOVELTY UPON EXISTING WORK

We note that ideas similar to our statistical top-$k$ have appeared in the literature. In particular, Shi et al. (2019) introduced the idea of fitting a Gaussian distribution to the entries of an input vector

---

[2] Total training cost relative to the base model. For finetuning based approaches, such as ReLUification (on Falcon and Llama) and ProSparse, the total training cost includes both the pretraining cost and the finetuning cost.

[3] Quality loss relative to the base model. Here the numbers are based on the results reported in their respective papers. Note that a different set of evaluation benchmarks is used in each paper. For ReLUification, this set contains ARC-Easy, HellaSwag, Lambada (for OPT) and Arc-E, Arc-C, Hellaswag, BoolQ, PIQA, LAMBADA, TriviaQA, WinoGrande, SciQ (for Falcon and Llama). For ProSparse, this set contains HumanEval, MBPP, PIQA, SIQA, HellaSwag, WinoGrande, COPA, BoolQ, LAMBADA, and TyDiQA. For HiRE, this set contains WMT14/WMT22, SuperGLUE, Multiple QA datasets, and multiple discriminative tasks datasets. For CATS, this set contains OpenBookQA, ARC Easy, Winogrande, HellaSwag, ARC Challenge, PIQA, BoolQ, and SCI-Q. For Spark Transformer, the datasets are those reported in Table 2.

[4] Results reported here are for the stage 1 training of their paper.

[5] Specifically, -80% on odd layers only, and -60% on average.

Table 4: Comparison with related work on enforcing activation sparsity in FFN of LLMs. Spark Transformer has the largest FLOPs reduction with one of the smallest quality loss.

| | Main Techniques | | | Main Results | | | |
|---|---|---|---|---|---|---|---|
| | Enforce sparsity | Predict support | Base model | Training cost[2] | Sparsity (%zeros) | Quality[3] | FFN FLOPs |
| ReLUification[4] (Mirzadeh et al., 2023) | ReLU | None | OPT 1.3B | +0% | 93% | -2% | -62% |
| | ReLU | None | Falcon 7B | +2% | 94% | -2.5% | -62% |
| | | | Llama 7B | +3% | 62% | -1.9% | -42% |
| ProSparse (Song et al., 2024a) | ReLU + $\|\cdot\|_1$ | None | Llama2 7B | +1.8% | 88% | -1.1% | -59% |
| | | | Llama2 13B | +6.7% | 88% | -1.4% | -59% |
| | | 2-layer FFN | Llama2 7B | NA | 75% | NA | NA |
| | | | Llama2 13B | NA | 78% | NA | NA |
| HiRE (Yerram et al., 2024) | Group top$_k$ + commonpath | Low-rank / quantization | PALM2 1B | +0% | 80% | -0.8% | -60%[5] |
| CATS (Lee et al., 2024b) | Thresholding | None | Mistral 7B | +0% | 50% | -1.5% | -33% |
| | | | Llama2 7B | | 50% | -2.4% | -33% |
| Spark Transformer | Statistical-top$_k$ | Partial dimensions | Gemma 2B | +0% | 92% | -0.9% | -72% |

and estimating a threshold from quantile functions. Then, M Abdelmoniem et al. (2021) extended the approach to additional distributions. In both cases, the method is used for solving the problem of distributed training. Here, we summarize our contribution upon these works:

- We are the first to use statistical top-$k$ for enforcing activation sparsity in Transformers. Improving Transformer efficiency via activation sparsity has become a very popular research topic (see Section 5), but may have been suffering from a lack of efficient top-$k$ algorithms for enforcing sparsity. Hence, the introduction of statistical top-$k$ may facilitate the development of this area.

- Synergizing statistical top-$k$ into Transformers is nontrivial. Since the method of statistical top-$k$ is based on fitting a statistical distribution to the activation vector, there is the need to understand the distributions of different activations in order to determine which particular activation vector is suited for the application of statistical top-$k$ and the associated choice of the distribution. In our case, we decide that statistical top-$k$ should be applied to the activation before the nonlinear function (for FFN) and before softmax (for Attention) since entries of this vector provably follows a Gaussian distribution at random initialization. We also verify empirically that statistical top-$k$ is still reliable even after initialization.

- We extended statistical top-$k$ from using the hard-thresholding operator with the estimated statistical threshold to the soft-thresholding operator. This leads to a continuous optimization landscape that may have facilitated the optimization. Empirically, we found this choice to provide quality benefits for Spark Transformer.

- We provide the first theoretical justification for the correctness of statistical top-$k$, see Theorem 1.

- We reveal the conceptual connection between statistical top-$k$ and several related top-$k$ operators in the literature, see Section 2.3. Such connections may motivate the development of more powerful top-k algorithms in the future.

## F.2 HANDLING CASES WHEN THE ACTIVATION IS SHARDED

The training of modern large Transformer models usually requires sharding certain model weights and activations across multiple computation devices, due to physical limitations on each device's memory. In particular, if sharding is used for the activations upon which the statistical top-$k$ is applied to, i.e., the pre-GELU activation in FFN and the pre-softmax activation in attention, extra care needs to be taken so that statistical top-$k$ is applied correctly. While this has not been the case

for our experiment on Gemma-2 2B, here we discuss possible solutions if this case arises, e.g., when training a larger Spark Transformer for which sharding relevant activations may become necessary.

Assume that an activation vector of length $N$ is sharded over $m$ devices, and we are interested in finding approximately the top-$k$ entries of $N$ with the largest value. There are two ways of applying statistical top-$k$ for this purpose.

- Global statistical top-$k$. Here we require each device to compute the mean and variance for entries on itself, then communicate them to all other devices. In this case, each device receives $m-1$ mean and variance values, which can be combined with mean and variance on its own to obtain global mean and global variance. Then, the rest of the steps in statistical top-$k$ can be carried out on individual devices. In this method, the output is exactly the same as if applying statistical top-$k$ without activation sharding. In terms of cost, there is extra computation coming from aggregating mean and variance from all devices, but the cost is very low as it requires only $O(k)$ FLOPs. The method also introduces a communication cost, but the cost is again small as each device only needs to send / receive $2k-2$ floating point numbers.

- Local statistical top-$k$. Here we simply apply statistical top-$k'$ to entries on its own device with $k' = k/m$. The method is sub-optimal in the sense that it does not necessarily produce the same output as applying the global statistical top-$k$. However, it has the benefit of not adding any computation and communication cost.

In cases where $k \ll N$, the global statistical top-$k$ above is cheap enough and hence could be a natural choice.

