# OpenReview forum: "Spark Transformer: How Many FLOPs is a Token Worth?"
_ICLR.cc/2025/Conference — Submitted to ICLR 2025_

### Official Review · Reviewer_1vWh · 2024-10-21

**Soundness:** 2
**Presentation:** 3
**Contribution:** 2
**Rating:** 5
**Confidence:** 4

**Summary:**

This paper employs an approximate statistical top-k algorithm, using the Gaussian quantile function to estimate the top-k threshold by calculating the mean and standard deviation of the activation maps. The statistical top-k algorithm is applied in sparse attention and integrated into the Gemma-2 model. Experiments were conducted, and the results demonstrate a speed-up compared to the original model.

**Strengths:**

This paper uses an approximate statistical top-k algorithm, leveraging the Gaussian quantile function to estimate the top-k threshold by calculating the mean and standard deviation of the activation maps. This approach requires only the computation of the mean and standard deviation to efficiently complete the selection, eliminating the time-consuming top-k selection process. Therefore, it is reasonable at least from the perspective of improving computational speed.

**Weaknesses:**

1. Only a theoretical estimation of the error for the approximate statistical top-k algorithm was provided, without experimental validation.
2. There was no analysis or testing of the gap between the approximate statistical top-k algorithm and the actual top-k selection.
3. The study lacks comparisons with a sufficient number of state-of-the-art sparse attention works.
4. Experiments were conducted only on CPUs, with no results provided for mainstream platforms such as GPUs.
5. There were no performance impact studies or experiments on applying the approximate algorithm.

**Questions:**

1. What is the actual distribution of the scores? How does it deviate from the Gaussian distribution?
2. What is the overlap between the selection results of the approximate statistical top-k and the actual top-k selection? How large is the error? What impact does it have on performance?

---

> ### Author Response · Authors · 2024-11-22
>
> Thank you for taking the time to read and review our paper!
>
>
> > Re: Only a theoretical estimation of the error for the approximate statistical top-k algorithm was provided, without experimental validation.
>
>
> We appreciate the reviewer's attention to detail. While we provided a theoretical analysis of the error for the approximate statistical top-k algorithm, we also included experimental validation in the original submission.
>
>
> Figure 1 demonstrates that the percentage of nonzeros resulting from statistical top-k in Spark Transformer closely aligns with the desired sparsity ratio. Our hyperparameter choices aimed for 8% nonzeros in Spark FFN (Figure 1(a)) and at most 256 nonzeros in Spark Attention (Figure 1(b)), which are accurately reflected in the empirical results.
>
>
> To further address the reviewer's comment, we have added Figure 6 and Figure 7 in the revised manuscript. These figures illustrate the error between the estimated threshold and the true threshold, demonstrating that the estimated threshold (red dotted vertical line) closely approximates the true threshold (blue vertical line). This observation strongly supports the effectiveness and accuracy of the statistical top-k algorithm in practice.
>
>
> > Re: There was no analysis or testing of the gap between the approximate statistical top-k algorithm and the actual top-k selection.
>
>
>
>
> We appreciate the reviewer's inquiry and would like to clarify the potential "gap" referred to in the question.
>
>
> -   **Gap in the number of nonzeros:** If the reviewer is referring to the difference between the number of nonzeros selected by statistical top-k and the desired sparsity level (k), we would like to direct their attention to Figure 1. This figure demonstrates a close match between the achieved sparsity and the target sparsity.
>
>
> -   **Gap in the threshold:** If the "gap" refers to the difference between the estimated threshold used in statistical top-k and the true threshold that would select exactly the top-k entries, we have added Figures 6 and 7 to the revised manuscript. These figures illustrate that the estimated threshold closely approximates the true threshold.
>
>
> -   **Gap in model quality:**  If the reviewer is concerned about the potential difference in model quality between using statistical top-k and exact top-k, we acknowledge the computational challenges in conducting a direct comparison. Training with exact top-k is prohibitively expensive (>10x slower), making this ablation study infeasible within the given timeframe. However, the strong agreement between statistical top-k and the desired sparsity level (Figure 1) and the accurate threshold approximation (Figures 6 and 7) suggest that statistical top-k serves as an effective proxy for exact top-k, minimizing any potential quality loss.
>
>
>
>
>
>
> > Re: The study lacks comparisons with a sufficient number of state-of-the-art sparse attention works.
>
>
> We appreciate the reviewer's suggestion. While we acknowledge the breadth of research in sparse attention, our work focuses on achieving sparsity through a different mechanism, namely, by inducing activation sparsity within the feedforward network. This approach offers a simpler alternative to complex sparse attention mechanisms, allowing for significant FLOPs reduction without intricate design modifications. We have included a discussion of related sparse attention works in the Related Work section to highlight the broader context of our contribution.
>
>
> > Re: Experiments were conducted only on CPUs, with no results provided for mainstream platforms such as GPUs.
>
>
> We appreciate the reviewer's observation. While our experiments focused on CPU evaluation, we believe that Spark Transformer can be effectively implemented on GPUs to achieve significant speedups. Recent work like Deja Vu and PowerInfer have demonstrated that dynamic, unstructured activation sparsity, similar to that in Spark Transformer, can be efficiently implemented on GPUs for Transformer architectures, leading to considerable performance gains.  We anticipate similar benefits for Spark Transformer given the shared sparsity characteristics. However, our primary focus was on addressing the modeling challenge of achieving high activation sparsity in LLMs without sacrificing quality. As demonstrated in Table 4, Spark Transformer attains higher activation sparsity with lower quality loss compared to existing methods. This contribution lays the groundwork for future efficient GPU implementations, building upon the aforementioned advancements in sparse GPU acceleration.

---

> > ### Comment · Reviewer_1vWh · 2024-11-29
> >
> > Thank you for addressing my questions. This paper now provides more detailed theoretical and experimental comparisons. However, as a mainstream computing platform, GPUs are not particularly friendly to sparsity, and effectively leveraging sparsity remains a challenging task. Simply stating the reduction in FLOPs is not sufficient to convince readers of the practicality of this work on GPU platforms. That said, based on the theoretical contributions of this paper, I will improve my score.

---

> > > ### Author Response · Authors · 2024-11-29
> > >
> > > We sincerely thank the reviewer for their thoughtful follow-up and for acknowledging the improvements in our revisions. We appreciate the constructive feedback and the updated assessment of our work.
> > >
> > > > Re: Practicality of Sparsity on GPUs
> > >
> > > We agree with the reviewer’s observation that GPUs pose challenges for effectively leveraging sparsity. However, we would like to emphasize that demonstrating runtime speedups on GPUs is not a universal requirement for every efficiency-related research. For example, influential works such as ReLU Strikes Back have been recognized for their theoretical and algorithmic contributions without presenting practical speedup results. Similarly, the core contribution of our work lies in achieving significant FLOP reductions while maintaining model quality, as shown in Table 4. This is a key step toward achieving more efficient LLMs, and we believe it makes an important contribution to the field.
> > >
> > > Additionally, the idea that activation sparsity yields runtime speedups on GPUs is already well-established in prior works like Deja Vu and PowerInfer. These studies demonstrate that sparsity mechanisms similar to those used in Spark Transformer can be effectively implemented on GPUs to achieve substantial runtime reductions. Including GPU benchmarks in our work would likely have reiterated these known findings rather than contributing new insights. Instead, our focus was to advance the modeling side of sparsity—achieving high levels of sparsity while preserving task performance—thereby addressing a critical gap in the existing literature.
> > >
> > > Lastly, while we deeply value the reviewer’s emphasis on demonstrating performance improvements on today's most popular hardware, we would also like to highlight the broader challenge of the [Hardware Lottery](https://arxiv.org/abs/2009.06489). This concept underscores the risk that innovative research ideas may be undervalued simply because they are not immediately suited to the prevailing software and hardware ecosystems, rather than due to any inherent limitations of the ideas themselves. As we discuss in our paper, we believe our work opens new avenues for the development of alternative hardware and platforms better optimized for sparse computations, paving the way for even greater efficiency gains in the future.
> > >
> > > Once again, we thank the reviewer for their invaluable feedback and their recognition of the contributions made in this work.

---

> > > > ### Comment · Reviewer_1vWh · 2024-11-30
> > > >
> > > > I remain cautious about the combination of GPUs and sparsity. Deja Vu and PowerInfer are successful works, and the most critical factor behind their success is the availability of open-source implementations and demonstrated real-world acceleration. I have seen many works claim FLOPs reduction for neural networks but offer no practical value. For a work discussing transformer acceleration, the lack of GPU timing results fails to convincingly demonstrate its practical impact.

---

> ### Author Response · Authors · 2024-11-22
>
> > Re: There were no performance impact studies or experiments on applying the approximate algorithm.
>
>
> We appreciate the reviewer's interest in the performance impact of the approximate statistical top-k algorithm. While a direct comparison with exact top-k would be ideal, it's important to highlight that training with exact top-k is computationally prohibitive (>10x slower), making this ablation study infeasible within the given timeframe.
>
>
> However, we provide strong evidence that statistical top-k closely approximates exact top-k. Figures 6 and 7 illustrate the minimal error between the estimated threshold used in statistical top-k and the true threshold for exact top-k selection. The close alignment of these thresholds indicates that statistical top-k selects nearly identical entries as exact top-k, minimizing any potential performance difference.
>
>
>
>
>
>
> > Re: What is the actual distribution of the scores? How does it deviate from the Gaussian distribution?
>
>
> We appreciate the reviewer's insightful question. To address this, we have added Figure 6 and Figure 7 in the revised manuscript, which illustrate the actual distribution of scores (blue curve) alongside a fitted Gaussian distribution (red dotted curve). As seen in the figures, the actual distribution closely matches the Gaussian distribution. While some minor discrepancies may exist in certain cases, the estimated threshold still closely aligns with the true threshold. This is because the deviations from the Gaussian distribution tend to cancel each other out, resulting in a sparsity level that closely matches the desired level, as evidenced in Figure 1.
>
>
>
>
> > Re: What is the overlap between the selection results of the approximate statistical top-k and the actual top-k selection? How large is the error? What impact does it have on performance?
>
> We appreciate the reviewer's insightful questions regarding the overlap and error between statistical top-k and exact top-k selection.
>
>
> To clarify the overlap:
> * If statistical top-k underestimates the selection threshold, it will include all the actual top-k entries plus some additional ones.
> * Conversely, if it overestimates the threshold, it will select the largest values among the actual top-k entries but miss some of the smaller ones.
>
>
> Regarding the error:
> * Figure 1 illustrates the error by comparing the actual number of nonzeros resulting from statistical top-k with the desired number of nonzeros. This provides a direct visualization of the accuracy of our approximation.
>
>
> While quantifying the precise performance impact of statistical top-k is challenging due to the prohibitive cost of training with exact top-k (>10x slower), Figures 1, 6, and 7 collectively demonstrate that statistical top-k serves as an effective approximation. The close alignment between the achieved sparsity and the desired sparsity, along with the accurate threshold estimation, suggests that any performance difference is likely to be minimal.

---

> ### Author Response · Authors · 2024-11-26
> **Gentle reminder**
>
> Dear Reviewer 1vWh,
>
> Thank you again for your valuable feedback, which has significantly improved our submission. In response to your comments, we have added Figure 6 and Figure 7 in the Appendix, which provides additional justifications for the effectiveness of our statistical top-k.
>
> As the discussion phase is ending soon, we wanted to kindly follow up to confirm if our response has addressed your concerns. Please don't hesitate to let us know and we are happy to answer any additional questions or address any concerns you may have.
>
> Thank you for your time and thoughtful input.
>
> Best regards,
> The Authors

---

> ### Author Response · Authors · 2024-11-30
>
> We thank the reviewer for their thoughtful follow-up and appreciate the continued dialogue on the practical impact of sparsity.
>
> > Re: Success of Deja Vu and PowerInfer due to open-source implementations and demonstrated acceleration
>
> We completely agree that real-world acceleration is critical for the success of works like Deja Vu and PowerInfer, as their primary focus is on hardware optimization for GPUs. However, these methods are not applicable to modern Transformers with non-ReLU activations, which lack activation sparsity. Our contribution addresses this gap by introducing a modeling approach that induces strong activation sparsity while preserving task performance. The success of our method in comparison to related work is directly measurable by the degree of sparsity achieved with minimal quality loss, as demonstrated in Table 4.
>
> > Re: FLOPs reduction claims without practical value
>
> We share the reviewer’s concern that not all forms of FLOPs reduction translate to practical value on GPUs. However, activation sparsity has repeatedly been shown in recent works, such as Deja Vu and PowerInfer, to directly lead to wall-time reductions as sparsity increases. Thus, maximizing FLOPs reduction through activation sparsity remains a justified and practical goal, with its potential for GPU speedups already validated in prior studies.
>
> **Summary**
>
> We would like to summarize why the lack of GPU-specific speedup results does not diminish the contribution of our work:
>
> 1. **Core Contribution**: The primary focus of our work is on modeling innovations to induce activation sparsity, not on system optimization.
>
> 2. **Established Practicality**: Prior works like Deja Vu and PowerInfer have already demonstrated that activation sparsity translates to practical GPU speedups, validating the relevance of our sparsity-focused approach.
>
> 3. **CPU Speedups**: We have shown practical acceleration on CPUs, providing further evidence of the real-world benefits of activation sparsity.
>
> 4. **Broader Impact**: Activation sparsity has implications beyond efficiency, informing theoretical studies on generalizability, learnability, and interpretability, as discussed in Section 5.
>
> 5. **Avoiding the Hardware Lottery**: Relying solely on results for today's hardware risks undervaluing ideas that could shape future hardware and software ecosystems.
>
> We thank the reviewer once again for their invaluable feedback, which has significantly enriched the discussion of our work.

---

> > ### Comment · Reviewer_1vWh · 2024-12-01
> >
> > Approximation and introducing sparsity do not improve the model's quality, so the goal of this work is acceleration rather than model innovation. If the GPU acceleration is as straightforward as you describe, showing me the GPU test results (CPU results are not enough) would be the most convincing way to fully gain my support for your work.

---

> > > ### Author Response · Authors · 2024-12-02
> > >
> > > Dear Reviewer,
> > >
> > > During the rebuttal period, we have worked diligently to explore the possibility of implementing custom GPU kernels to demonstrate wall-time benefits.
> > >
> > > Please note that we never claimed GPU acceleration to be “straightforward.” On the contrary, achieving acceleration on GPUs requires significant system-level expertise, as demonstrated by dedicated efforts like Deja Vu and PowerInfer. Our intention has been to highlight that such established results already exist, and replicating them in our submission would not substantially enhance the novelty of our work, which centers on modeling innovations to induce activation sparsity hence complementing Deja Vu / PowerInfer etc.
> > >
> > > We will continue and try our best to have GPU implementation given the reviewers interest, but may not be able to make it before the end of the rebuttal period. Meanwhile, we would be happy to address any remaining question the reviewer may have.
> > >
> > > Best

---

### Official Review · Reviewer_znbP · 2024-10-28

**Soundness:** 3
**Presentation:** 3
**Contribution:** 2
**Rating:** 6
**Confidence:** 4

**Summary:**

Sparse transformer is an approach to select statistical top k entries from the input with two goals

1. To reduce computation of entries not in top k
2. To achieve the same in an efficient manner  (complexity - 2d vs O(dlogd))
3. To get speedup/Lower FLOP for efficient inference (on CPU only)

The above goals are quantified by showcasing the results where by using 8% non-zeros in the FFN activation and attending to a maximum of 256 tokens, the model achieves 3.1× reduction in FLOPS resulting in 1.70× speedup for preﬁll and a 1.79× speedup for decoding on a 16-core CPU VM. No speedup is shown on other devices like GPU, accelerators because inference needs special kernel which is available in Gemm-2 implementation.

**Strengths:**

- Solving an important problem of transformers (increasing speed by reducing the number of FLOPS)
- Mathematical explanation of statistical top k entry is solid and clear to readers. Both Spark FNN and Spark Attention approach makes sense and expected to reduce FLOPs (with certain impact on accuracy)
- Showing the FLOP reduction using a custom matrix multiplication (sparse vector matrix multiplication) which gives further boost to theoretical explanation of FLOP reduction.

Overall a good paper proposing a new method of exploiting fine-grain sparsity in transformer.

**Weaknesses:**

- Although the method is convincing and shows speedup on CPU using special implementation of GEMM (sparse GEMM), the Author(s) need to show approaches which can convince readers that this method can potentially be useful for mainstream hardware like GPUs, custom accelerators. [Semi-structured sparsity]((https://arxiv.org/abs/2309.06626/) and [Deep Adaptive Inference Networks](https://arxiv.org/abs/2004.03915) methods also proposed similar approach for CNN but has similar limitation of speedup being shown only for CPUs.
- L324 : The comparison with state of the art methods like ProSparse can be done using same model so that we can see a fair comparison. The table suggests that for first four benchmark, ProSparse and LLaMMA ReGLU perform better (lower FLOP/token)
- Comparison with token pruning methods and impact on inference of end to end model with Sparse transformer will strengthen this paper.

My rating is based on the fact that this method may not be able to scale to other hardware and speedup on CPU alone is not significant. Even if it works for other Edge hardware, it might be a good contribution.

**Questions:**

- L065: $0.5*d_{model} * d_{ff} + 1.5 * d_{model} * k_{ff}$ , Authors are encourage to explain the origin of factors 0.5 and 1.5.
- Authors are also requested to provide a comparison with other token pruning methods to compare speedup with state of the art methods ? it will be helpful to amplify the contribution of this paper.
- What is the future plan to make this method work on other hardware like GPU/Accelerators ? Making this method work for other hardware will increase the usefulness of this approach.

---

> ### Author Response · Authors · 2024-11-22
>
> Thank you for taking the time to read and review our paper!
>
>
> > Re: Speedup on GPU / TPUs
>
>
> While our experiments focused on CPU evaluation, we believe that Spark Transformer can be effectively implemented on GPUs to achieve significant speedups. Recent work like Deja Vu and PowerInfer have demonstrated that dynamic, unstructured activation sparsity, similar to that in Spark Transformer, can be efficiently implemented on GPUs for Transformer architectures, leading to considerable performance gains.  We anticipate similar benefits for Spark Transformer given the shared sparsity characteristics. However, our primary focus was on addressing the modeling challenge of achieving high activation sparsity in LLMs without sacrificing quality. As demonstrated in Table 4, Spark Transformer attains higher activation sparsity with lower quality loss compared to existing methods. This contribution lays the groundwork for future efficient GPU implementations, building upon the aforementioned advancements in sparse GPU acceleration.
>
>
> > Re: Using the same model as ProSparse for a fair comparison
>
>
>
>
> We acknowledge that comparing with ProSparse/ReGLU using the same baseline model would be ideal. However, ProSparse/ReGLU focus on fine-tuning existing models, while our method introduces a new architecture designed for pre-training from scratch. This fundamental difference makes a direct comparison on Llama2-7B challenging, as the pre-training data required to train a Spark Transformer version of Llama2-7B is not publicly available.
>
>
>
>
> > Re: Comparison to token pruning
>
>
> We have added a short discussion of KV cache pruning methods, such as Heavy-hitter oracle and Scissorhands, to the related work section. These methods drop less-used tokens from the KV cache during decoding. This approach has the drawback that dropped tokens are permanently lost, even if they become useful later in the decoding process. This limitation may explain why these methods only achieve a 5x compression rate. In contrast, our method selectively retrieves information from the entire KV cache. Notably, we showed that Top-256 retrieval is sufficient regardless of the context window length, enabling a much higher compression rate with larger contexts.
>
>
> > Re: explaining L065
>
>
> We appreciate the reviewer's attention to detail. L065 is explained in the paragraph "FLOPS per Token" in Section 3.1. We have added a pointer to this explanation in footnote 1 in the revised manuscript to clarify how L065 is obtained.

---

> > ### Comment · Reviewer_znbP · 2024-11-23
> >
> > Appreciate Authors for providing updated version of the paper as well as explanation of all the questions.
> >
> > I would keep my rating same given the limitation of this method for CPU only as of now (GPU/accelerator results might make this paper stronger especially since this paper talks about efficiency).

---

> > > ### Author Response · Authors · 2024-11-23
> > >
> > > We sincerely thank the reviewer for taking the time to read our rebuttal. We understand the concern regarding the lack of GPU/accelerator results in our work, especially in the context of efficiency.
> > >
> > > However, we would like to emphasize that demonstrating runtime speedups on GPUs or accelerators is not a necessary requirement for all efficiency-related papers. For example, the [ReLU Strikes Back](https://openreview.net/forum?id=osoWxY8q2E) paper was considered impactful despite not presenting any practical speedup results, as it focused on the theoretical and algorithmic aspects of model efficiency. Similarly, in our work, the primary contribution lies in demonstrating a significant reduction in FLOPs without sacrificing model quality, as shown in Table 4. This is a key step toward achieving more efficient LLMs, and we believe it makes an important contribution to the field.
> > >
> > > Moreover, the idea that activation sparsity can lead to speedups on GPUs is well-established in recent works such as Deja Vu and PowerInfer. Therefore, even if we had included GPU results in our submission, they would have largely reiterated known findings, rather than contributing new insights. Our focus on minimizing FLOPs without quality degradation is the novel and crucial aspect of this paper, which has yet to be fully addressed in the literature.
> > >
> > > We hope this clarifies our position and we appreciate the reviewer’s consideration of the broader impact of our work.

---

> > > > ### Comment · Reviewer_znbP · 2024-11-23
> > > >
> > > > Thank you for providing further insights to answer my concerns/questions. [ReLU Strikes Back](https://openreview.net/forum?id=iOy2pITOoH&referrer=%5BReviewers%20Console%5D(%2Fgroup%3Fid%3DICLR.cc%2F2025%2FConference%2FReviewers%23assigned-submissions)#:~:text=For%20example%2C%20the-,ReLU%20Strikes%20Back,-paper%20was%20considered) paper not only reduces the FLOP but indirectly induces more sparsity so it is more than just FLOP reduction. FLOP reduction is an indicator of efficiency improvement but it is not a guarantee because different hardware architecture and compilers has different impact on these techniques. So, although, I am convinced with the contribution of the paper but it will be better to show the improvement on actual hardware (with different architecture). I will keep my score.

---

> > > > > ### Author Response · Authors · 2024-11-24
> > > > >
> > > > > We sincerely thank the reviewer for their continued engagement and valuable feedback.
> > > > >
> > > > > The reviewer is absolutely correct that FLOP reduction does not necessarily guarantee runtime reduction. Importantly, the FLOP reduction achieved in our Spark Transformer stems directly from enforcing activation sparsity—same as the mechanism highlighted in ReLU Strikes Back. In fact, the central objective of our work is to maximize activation sparsity while preserving model quality, thereby achieving substantial FLOP reduction. As demonstrated in related works such as Deja Vu and PowerInfer, this type of sparsity is highly likely to translate into GPU runtime reductions, which further supports the practical potential of our approach.
> > > > >
> > > > > Lastly, while we deeply value the reviewer’s emphasis on demonstrating performance improvements on today's most popular hardware, we would also like to highlight the broader challenge of the [Hardware Lottery](https://arxiv.org/abs/2009.06489). This concept underscores the risk that innovative research ideas may be undervalued simply because they are not immediately suited to the prevailing software and hardware ecosystems, rather than due to any inherent limitations of the ideas themselves. As we discuss in our paper, we believe our work opens new avenues for the development of alternative hardware and platforms better optimized for sparse computations, paving the way for even greater efficiency gains in the future.
> > > > >
> > > > > Once again, we thank the reviewer for their thoughtful and constructive critique, which has enriched the discussion and helped to situate our contributions in a broader context.

---

> > > > > > ### Comment · Reviewer_znbP · 2024-11-26
> > > > > > **Score revision**
> > > > > >
> > > > > > Thank you for providing more insights. Based on all these discussions, I am convinced to improve my score.

---

> > > > > > > ### Author Response · Authors · 2024-11-26
> > > > > > > **Thanks again for the thoughtful discussion**
> > > > > > >
> > > > > > > Dear Reviewer znbP,
> > > > > > >
> > > > > > > We sincerely appreciate your thoughtful engagement in the discussion. We will make sure to improve our manuscript to incorporate the valuable points you have raised.
> > > > > > >
> > > > > > > Best,
> > > > > > > Authors

---

### Official Review · Reviewer_8pJV · 2024-10-31

**Soundness:** 3
**Presentation:** 3
**Contribution:** 2
**Rating:** 6
**Confidence:** 3

**Summary:**

This paper proposes a statistical top k algorithm and a low rank predictor as a sparsity regularizer during pretraning, which maintains similar model quality benchmark as the dense counterpart (aka GEMM 2) and achieves improved (e..g, ~2X) inference speed up on CPU (or SIMD machines).

**Strengths:**

1. The proposed statistical top K method is intuitive and algorithm/hardware efficient
2. Both of the Sparisity regularizers (Statistical top K and low rank predictor) are applied during pretraining, thus saving post-training adapters and potentially contributes to close to SOTA model quality
3. analytical modeling is clear and shows it is an error-bound system
4. Due to the activation sparsity, it reduces the flops-heavy ops during inference on SIMD machines as table look-up instead of dense GEMMs thus demonstrates significat speed up on commercially more available CPUs

**Weaknesses:**

The table look up is efficient in SIMD machines but not friendly to accerlerators with Tensor cores or systolic arrays like GPU/TPUs. Though CPUs are much more commercially available than high end GPUs, it is unclear if the tokens/watt or tokens/second/watt of CPUs can beat GPUs during batched (instead of batch size 1) inference, even with sparsity

**Questions:**

1. It is unclear to me how important it is have the two regularizers in pre-train or post-train (e..g, how does it contribute to final model quality), if it is still effective with just post-training/fine-tune, then it is more applicable to wider range of models

2. why is the score of HumanEval of GEMMA  2 much lower than published ones in table 2?

3. Is the model trained with Tensor Parallel? If not, how does the statistical top K performs when the activation is sharded? Would it incur extra CCL overhead or skewed top-K distribution (over devices) that further slows down the training?

---

> ### Author Response · Authors · 2024-11-22
>
> Thank you so much for your time, insightful feedback, and the positive evaluation :-)
>
>
>
>
> > Re: table lookup is not friendly to accelerators
>
>
> We appreciate the reviewer's observation. While our experiments focused on CPU evaluation, we believe that Spark Transformer can be effectively implemented on GPUs to achieve significant speedups. Recent work like Deja Vu and PowerInfer have demonstrated that dynamic, unstructured activation sparsity, similar to that in Spark Transformer, can be efficiently implemented on GPUs for Transformer architectures, leading to considerable performance gains.  We anticipate similar benefits for Spark Transformer given the shared sparsity characteristics. However, our primary focus was on addressing the modeling challenge of achieving high activation sparsity in LLMs without sacrificing quality. As demonstrated in Table 4, Spark Transformer attains higher activation sparsity with lower quality loss compared to existing methods. This contribution lays the groundwork for future efficient GPU implementations, building upon the aforementioned advancements in sparse GPU acceleration.
>
>
> > Re: how does each of the two techniques (top-k and predictor) contribute to final quality
>
>
>
>
> We appreciate the reviewer's insightful question. To address this directly, we have added additional ablation studies in Section D.4 of the revised manuscript. Our results demonstrate that applying top-k alone without the low-cost predictor leads to a decrease in model performance. This suggests that the low-cost predictor, while introduced primarily to reduce the FLOPs required for locating nonzero entries, also plays a crucial role in making the model more amenable to sparsification and maintaining high performance.
>
>
>
>
>
>
> > Re: are the two techniques effective for fine-tuning
>
>
> Our method requires architectural changes beyond simply switching the activation function as in many previous works, hence cannot be directly applied upon a pretrained model. We recognize that this may be seen as a shortcoming of Spark Transformer as it is not suited for cases where one is seeking to adapt an existing model to having activation sparsity. However, the goal of this paper is to demonstrate that Spark Transformer can be used as a drop-in replacement of the standard Transformer architecture (see line 082), and we advocate adopting Spark Transformer as the architecture in future pretrained models. In that case, the activation sparsity comes entirely for free from future models, which has a big advantage over sparse fine-tuning based approaches which not only incurs training complications but also cannot achieve the same level of FLOPs reduction as ours.
>
>
> > Re: HumanEval of Gemma 2 lower than published ones
>
>
> Thanks for noting this. We have reached out to the authors of Gemma 2 for understanding this discrepancy and we are looking to resolve it.
>
>
> > Re: statistical topK when activation is sharded
>
>
>
>
>
>
> The training of Spark Gemma-2 2B in our paper does not involve sharding the activations upon which statistical top-k is applied to.  However, we acknowledge that sharding activations is a valid practical concern and have added a discussion in Appendix F.2. Statistical top-k can be easily extended to this scenario with minimal computational overhead and device communication. Each device can compute the local mean and variance of its activation entries and then broadcast these two numbers to all other devices. This allows each device to efficiently compute the global mean and variance. The communication cost is minimal, as each device only needs to send and receive 2 * #devices floating point numbers per activation vector, which typically contains thousands of entries.

---

> > ### Comment · Reviewer_8pJV · 2024-11-23
> >
> > Thanks for addressing my questions. The discussion on topK sharding makes sense to me. I would keep my score.

---

> > > ### Author Response · Authors · 2024-11-23
> > >
> > > We appreciate the reviewer for taking time to read and respond to our rebuttal. We would be very happy to address any remaining concerns that the reviewer may have.

---

### Official Review · Reviewer_uU7z · 2024-11-03

**Soundness:** 2
**Presentation:** 2
**Contribution:** 1
**Rating:** 5
**Confidence:** 4

**Summary:**

From the trials to replacing non-sparsified activations (e.g., SiLU) to sparsified activations (e.g., ReLU), sparsity-based LLM inference acceleration is widely studied due to its effectiveness in reducing the computational cost. It generally adopts non-ReLU activation and sparsity predictors for acceleration. This paper, Spark Transformer, aims for the same target (i.e., exploiting sparsity) but introduces a sparsity-exploiting drop-in replacement structure for the transformer while providing an end-to-end exploitation of sparsity in LLM. From this replacement, Spark Transformer can reduce the post-processing overhead (to train sparsity predictors additionally) compared to existing non-ReLU approaches. For this alternative solution, Spark Transformer proposes statistical Top-K and Spark Attention, which utilizes statistical Top-K. When trained from scratch using Spark Transformer (based on Gemma), it achieves around 1.7-1.8x speedup compared to the original Gemma (not compared to a previous sparsity-based LLM inference acceleration) through this sparsity-exploiting structure.

**Strengths:**

1. Important Domain: This paper tackles a timely domain, sparsity-based LLM inference.

2. New Approach: This paper proposes a new structure for sparsity-based LLM inference. While previous works mainly focus on better estimating sparsity through predictors, this paper introduces a drop-in replacement structure for transformers. Also, it exploits the attention sparsity for both FFN and attention.

3. Profound Descriptions: This paper provides profound descriptions of its methods with proofs.

**Weaknesses:**

1) Limited novelty: While statistical Top-K could seem to be a fresh approach in the sparsity-based LLM inference, the Top-K methods, especially statistical Top-K [1], were already extensively researched areas in the gradient/activation compression [1,2,3]. As previous statistical Top-K gradient compression [1] introduced a profound statistical Top-K selection (through threshold estimation) for various statistical distributions, the paper should show the main differences and emphasize the novelty. ([2] introduced Gaussian-based threshold estimation, and [1] significantly extended it with various distributions.) Also, [6] seems to propose a similar approach for the partial weights to predict the approximate scores of tokens.

2) Practicality: 1. Does this method provide inference time speedup compared to existing sparsity-based LLM inference? (e.g., DejaVu [4,5,6]). The main speedup is compared to the original Gemma only in the evaluation section. 2. Is single-stage training more practical than preprocessing existing sparsity-based LLMs? The introduction mentions this as one of the paper's strong points, so it should be tested quantitatively or with experiments. 3. As far as the reviewer understands, Spark Transformer requires a pretraining from scratch. Is this choice preferable to the predictor overhead of existing sparsity-based LLM inference acceleration?

3) Insufficient evaluations: This paper provides an in-depth analysis of the methods in the main body, but the evaluations are somewhat limited. Accuracy analysis is limited, and speedup comparison is conducted only with the original model (not sparsity-based LLM inference acceleration baselines). Also, there are many top-K methods to address the top-K selection overhead (e.g., quasi-sort with O(logN) complexity [7]), so statistical Top-K should be compared with those works excluding JAX top-K implementation.

[1] An Efficient Statistical-based Gradient Compression Technique for Distributed Training Systems, MLSys, 2021.
[2] Understanding Top-k Sparsification in Distributed Deep Learning, arXiv 1911.08772, 2019.
[3] Deep Gradient Compression: Reducing the Communication Bandwidth for Distributed Training, ICLR, 2018.
[4] Deja Vu: Contextual Sparsity for Efficient LLMs at Inference Time, ICML, 2023.
[5] H2O: Heavy-Hitter Oracle for Efficient Generative Inference of Large Language Models, NeurIPS, 2024.
[6] InfiniGen: Efficient Generative Inference of Large Language Models with Dynamic KV Cache Management, OSDI, 2024.
[7] Scalecom: Scalable Sparsified Gradient Compression for Communication-Efficient Distributed Training, NeurIPS, 2019.

**Questions:**

I enjoyed reading this paper, which tackles the timely topic of sparsity-based LLM inference acceleration. The main worrisome aspects of this paper are two-fold: novelty and practicality. The following are my questions related to them.

(Novelty)

Q1: What is the main difference between this paper and the existing statistical Top-K methods ([1,2])? I think [1,2,3] should be considered as related works in the paper. Additionally, please compare Spark Transformer with [6], which proposes a similar approach for the partial weights to predict the approximate scores of tokens.

(Practicality)

Q2: How much speedup (e.g., tokens per second) does Spark Transformer provide compared to the existing sparsity-based LLM inference acceleration (e.g., DejaVu [4], ProSparse, LLaMa ReGLU)?

Q3: What is the actual benefit (e.g., in terms of training time) of reducing preprocessing overhead through Spark Transformer?

Q4: Please compare Spark Gemma-2 with other accuracies. While the current evaluation compares Spark Gemma-2 with MMLU, GSM8K, AGIEval, BBH, Windogrande, and HellaSwag, as Spark Transformer may affect accuracy, wider comparison should be conducted. [5] provides a simple and easily accessible API for evaluating accuracies, so it would not be that difficult to compare.

Q5: Most of the previous works tried to avoid pretraining costs, but the reviewer wonder why this work chooses to train the model from scratch, which takes a huge overhead to prove the real effect of this work.

[5] lm-evaluation-harness: https://github.com/EleutherAI/lm-evaluation-harness

(Other Points)

Q6. This paper only shows the experiments on the CPU, not any kind of GPUs or TPUs. Does Spark Transformer show similar speedup on GPUs or TPUs?

Q7. The reviewer guesses there is a typo on page 6, line 315. Eq.(13) should be changed to Eq.(14).

Here are some of the suggestions to make the paper more valuable.
- Novelty: Emphasizing the main difference between Sparse Top-K’s statistic al Top-K and the previous Top-K approaches would make the paper more novel. Providing the difficulty of synergizing the Top-K to the transformer-based structure would be a good option.

- Practicality: Providing more extensive evaluations to show the speedup and stable accuracy of Spark Transformer in various settings would significantly enhance its practicality. Also, please show that the pretraining of Spark Transformer is more efficient than the existing sparsity-based LLM inference acceleration.

---

> ### Author Response · Authors · 2024-11-22
>
> Thank you for taking the time to read and review our paper!
>
>
> > Re: Q1 - Novelty
>
>
>
>
>
>
> On novelty compared to [1, 2, 3]: We would like to thank the reviewer for bringing Refs [1, 2, 3] to our attention. While we acknowledge that they are related works, our work differs significantly by applying and adapting statistical top-k to activation sparsity within Transformer architectures. We elaborate on this point below.
>
>
> - We are the first to use statistical top-k for enforcing activation sparsity in Transformers. Improving Transformer efficiency via activation sparsity has become a very popular research topic (see Section 5), but may have been suffering from a lack of efficient top-k algorithms for enforcing sparsity. Hence, the introduction of statistical top-k may facilitate the development of this area.
>
>
> - Synergizing statistical top-k into Transformers is nontrivial. Since the method of statistical top-k is based on fitting a statistical distribution to the activation vector, it is necessary to understand the distributions of different activations in order to determine which particular activation vector is suited for the application of statistical top-k and the associated choice of distribution. In our case, we decided that statistical top-k should be applied to the activation before the nonlinear function (for FFN) and before softmax (for Attention) since entries of this vector provably follow a Gaussian distribution at random initialization. We also verify empirically that statistical top-k is still reliable even after initialization (see Figure 1).
>
>
> - We extend statistical top-k from using the hard-thresholding operator with the estimated statistical threshold to using the soft-thresholding operator. This leads to a continuous optimization landscape that may have facilitated the optimization. Empirically, we found this choice to provide quality benefits for Spark Transformer.
>
>
> - We provide the first theoretical justification for the correctness of statistical top-k; see Theorem 1.
>
>
> - We reveal the conceptual connection between statistical top-k and several related top-k operators in the literature; see Section 2.3. Such connections may motivate the development of more powerful top-k algorithms in the future.
>
> On novelty compared to [6]: We also thank the reviewer for pointing us to Ref. [6], which uses an idea similar to the reference (Yang et al., 2024) that we discussed in the introduction. Both [6] and Yang et al. (2024) use a subset of the key dimensions to estimate the important keys, which is similar to Spark Transformer. However, identifying those dimensions requires an arguably complex procedure, and those dimensions are input-dependent, so they need to be recomputed each time. In contrast, we simply use a predetermined set of dimensions that can be used for all inputs.
>
>
> In the revised manuscript, we have added discussions of Refs [1, 2, 3] on lines 103-105 and an extended discussion in Appendix F. We have also added a reference to Ref [6] on line 76.
>
>
>
>
>
>
>
>
> > Re: Q2 - Practically in terms of speedup over comparing method
>
>
> We appreciate the reviewer's question regarding the practical speedup of our method. It's important to recognize that the final speedup observed in practice is influenced by two key factors:
>
>
> 1.  **FLOPs reduction without sacrificing model quality:** This is the primary focus of our work. We introduce Spark Transformer as a novel architecture designed to maximize FLOPs reduction while maintaining model quality. To address the reviewer's comment on comparisons with related methods, we have added Table 4 in Appendix F, which lists the FLOPs reduction and quality loss of various approaches. This table demonstrates that Spark Transformer achieves state-of-the-art FLOPs reduction with minimal quality loss.
>
>
> 2.  **ML system implementation that effectively leverages sparsity:** While a comparison of the final speedup would be informative, it's important to note that such comparisons are difficult to make across different papers.  The final speedup is highly dependent on various factors such as the specific hardware used (CPU, GPU, TPU), hardware design choices (HBM vs. compute tradeoff, tensor core vs. sparse core), and the specific implementation details. These factors vary significantly across research works, making direct comparisons challenging.  In fact, none of the prior works in this area provide such a comparison for this very reason.
>
>
> FLOPs reduction, on the other hand, offers a fair and consistent comparison across methods that utilize the same form of activation sparsity (as in Table 4). This is because, given the same hardware and implementation, methods with larger FLOPs reduction are likely to exhibit larger speedups. Our focus on maximizing FLOPs reduction while preserving model quality provides a strong foundation for achieving significant speedups in practice when combined with optimized sparse implementations on various hardware platforms.

---

> > ### Author Response · Authors · 2024-11-22
> >
> > > Re: Q3 - practicality in terms of reducing preprocessing overhead
> >
> >
> > Thank you for raising this important point about practicality. To clarify the advantages of Spark Transformer's single-stage training and reduced preprocessing overhead, we have incorporated Table 4, which includes a "Training cost" column.
> >
> >
> > As this table highlights, several prior works, such as ReLUification and ProSparse, rely on fine-tuning. This approach necessitates both a pretrained model and an additional training cost ranging from 1.8% to 6.7% beyond the initial pretraining. In contrast, Spark Transformer eliminates this overhead by integrating sparsity directly into the training process. This means it incurs the same pretraining cost as the baseline model without any additional fine-tuning expenses. This streamlined approach offers a significant practical advantage in terms of both time and computational resources.
> >
> >
> >
> >
> > > Re: Q4 - practicality in terms of evaluating on additional benchmarks
> >
> >
> > We appreciate the reviewer's suggestion to evaluate on additional benchmarks. We would like to highlight that, in addition to the 6 benchmarks listed by the reviewer (MMLU, GSM8K, AGIEval, BBH, Windogrande, and HellaSwag), we evaluated Spark Transformer on 10 other benchmarks, as detailed in Table 2.
> >
> >
> > This comprehensive collection of 16 benchmarks allows us to assess various aspects of the pretrained model's capabilities, including commonsense reasoning, world knowledge, reading comprehension, math, and code.  It encompasses all datasets used to evaluate recent open LLM models like Gemma, Mistral, and Llama2. Furthermore, this collection is more extensive than the benchmarks used in prior work on sparse activation, as noted in the added footnote #3 in the paper.
> >
> >
> > Therefore, we believe that our evaluation convincingly demonstrates that Spark Transformer maintains strong performance across a diverse range of tasks without substantial quality loss.
> >
> >
> >
> >
> >
> >
> > > Re: Q5 - practicality in terms of comparing pretraining vs finetuning approaches
> >
> >
> >
> >
> > We appreciate the reviewer's question about the practicality of pretraining versus fine-tuning approaches with Spark Transformer. The primary goal of our paper is to demonstrate that Spark Transformer can serve as a drop-in replacement for the standard Transformer architecture. As stated in line 082, it achieves significant FLOPs reduction without introducing training complications or incurring quality loss.  Essentially, we advocate for adopting Spark Transformer as the default architecture for future pretrained models. This approach offers a major advantage over sparse fine-tuning methods: the activation sparsity in Spark Transformer comes entirely for free. In contrast, sparse fine-tuning methods not only add training complexity but also typically achieve lower levels of FLOPs reduction.
> >
> >
> > However, we acknowledge that if the objective is to obtain a sparsely activated model given existing pretrained models, there is a tradeoff between pretraining a new model from scratch using Spark Transformer and fine-tuning a pre-existing model. Pretraining with Spark Transformer offers the benefits of simplicity and greater FLOPs reduction, but it requires more computational resources.  Fine-tuning, on the other hand, may be faster but might not achieve the same level of sparsity and efficiency.
> >
> >
> >
> >
> >
> >
> > > Re: Q6 - Does Spark Transformer show similar speedup on GPUs or TPUs?
> >
> >
> > While our experiments focused on CPU evaluation, we believe that Spark Transformer can be effectively implemented on GPUs to achieve significant speedups. Recent work like Deja Vu and PowerInfer have demonstrated that dynamic, unstructured activation sparsity, similar to that in Spark Transformer, can be efficiently implemented on GPUs for Transformer architectures, leading to considerable performance gains.  We anticipate similar benefits for Spark Transformer given the shared sparsity characteristics. However, our primary focus was on addressing the modeling challenge of achieving high activation sparsity in LLMs without sacrificing quality. As demonstrated in Table 4, Spark Transformer attains higher activation sparsity with lower quality loss compared to existing methods. This contribution lays the groundwork for future efficient GPU implementations, building upon the aforementioned advancements in sparse GPU acceleration.
> >
> >
> > > Re: Q7 - typo in line 315, Eq.(13) should be changed to Eq.(14).
> >
> >
> > We appreciate the reviewer's careful reading and attention to detail.  The reference to Eq. (13) on line 315 is indeed correct. It is intentionally referring to the regular attention mechanism (as opposed to Spark Attention) used in the standard Gemma 2 architecture. We are happy to elaborate further if this clarification does not fully address the reviewer's concern.

---

> > > ### Author Response · Authors · 2024-11-22
> > >
> > > > Re: there are many top-K methods to address the top-K selection overhead (e.g., quasi-sort with O(logN) complexity [7])
> > >
> > >
> > >
> > > Thank you for raising this point about alternative top-k methods. While efficient top-k selection is an active research area, it's important to note that no prior work on activation sparsity has utilized any approximate top-k algorithm. Therefore, the introduction of statistical top-k itself represents a significant contribution to the field.
> > >
> > >
> > > We acknowledge the existence of other recently developed approximate top-k algorithms, such as soft top-k and SparseK, as discussed in Section 2.3. However, we did not include them in our experimental study for two primary reasons. First, they lack explicit mechanisms for controlling the sparsity level, a crucial requirement for our application. Second, these iterative algorithms are considerably more computationally expensive than our proposed method.
> > >
> > >
> > > Regarding Ref. [7], our understanding is that its top-k algorithm is specifically designed for distributed training scenarios. It is not a general-purpose approximate top-k algorithm readily applicable to activation sparsity.
> > >
> > >
> > >
> > >
> > >
> > >
> > > > Re: additional suggestions.
> > >
> > >
> > > We appreciate the reviewer's valuable suggestions and have incorporated them into the revised manuscript. Specifically, we have:
> > >
> > >
> > > 1.  Added a few sentences at the end of the introduction and included an extensive discussion in Appendix F to further clarify the novel contributions of our work.
> > >
> > >
> > > 2.  Added Table 4 to provide a detailed comparison with related works in terms of FLOPs reduction, training cost, and model quality.

---

> ### Author Response · Authors · 2024-11-26
> **Gentle reminder**
>
> Dear Reviewer uU7z,
>
> Thank you again for your valuable feedback, which has significantly improved our submission. In response to your comments, we have added several discussions and experimental results in the Appendix, highlighted in red in the revised manuscript.
>
> As the discussion phase is ending soon, we wanted to kindly follow up to confirm if our response has addressed your concerns. Please don't hesitate to let us know and we are happy to answer any additional questions or address any concerns you may have.
>
> Thank you for your time and thoughtful input.
>
> Best regards,
> The Authors

---

> > ### Comment · Reviewer_uU7z · 2024-11-27
> >
> > I thank the authors for the response and the rebuttal. I have read the rebuttal and discussion with other reviewers.
> > Some of my concerns are addressed, but still many are unresolved, along with a few more newly arising.
> >
> > - On the novelty, the current version seems to give adequate credit to previous work. What's remaining as a contribution seems slightly weak, but I partially agree that it's meaningful.
> >
> > - What I cannot easily agree is that the authors are claiming Spark Transformer to be the default architecture.
> > This seems to be too aggressive, and does not align well with what's said in the paper. I don't think it will be easy to use this as the default, unless there is a very easy way to find the optimal $k$, with at least semi-guarantee that it's not going to lose final accuracy. For such a purpose, I believe it requires more backup methods and much more evaluation.
> >
> > - According to the rebuttal, and especially since the authors have mentioned the hardware lottery, I believe the proposed method has to be compared with other existing methods in the field of unstructured pruning. In fact, this aligns with my original question Q4. What I was asking was to add more baselines and/or fill in the empty cells such that the readers can see if the proposed method indeed achieves SOTA performance in an apples to apples comparison. The authors seem to have taken this as a request for adding more benchmarks.
> >
> > - On the alternative scenario where a pretrained model is given (which I believe to be more likely), the tradeoff between training time vs flops reduction and accuracy has to be quantified. I believe this brings us back to Q4, where a fair accuracy comparison should be possible.

---

> > > ### Author Response · Authors · 2024-11-29
> > >
> > > We sincerely thank the reviewer for their detailed follow-up and constructive feedback. Below, we address the remaining concerns.
> > >
> > > > Re: Spark Transformer as the Default Architecture
> > >
> > > We acknowledge the reviewer’s concerns about the claim of Spark Transformer being the default architecture. Our intent was to emphasize Spark Transformer’s potential as a compelling alternative architecture, particularly for cases where sparsity in activation is desired. We believe this assertion is well-supported by our results and the following considerations:
> > >
> > > *Seamless Integration Without Additional Overhead.* Spark Transformer achieves activation sparsity at no extra training cost, as it integrates sparsity directly into the pretraining process. Unlike finetuning-based methods that require a pretrained model and additional processing, Spark Transformer eliminates this overhead, making it more practical. This simplicity aligns with the requirements for an "alternative" architecture that can be broadly adopted without specialized preprocessing or training modifications.
> > >
> > > *Strong Empirical Results Across Multiple Benchmarks.* Our extensive evaluations on 16 diverse benchmarks demonstrate that Spark Transformer maintains strong task performance. This comprehensive suite of benchmarks allows us to assess various aspects of the pretrained model's capabilities, including commonsense reasoning, world knowledge, reading comprehension, math, and code. It encompasses all datasets used to evaluate recent open LLM models such as Gemma, Mistral, and Llama2. Such consistent results suggest that Spark Transformer is a competitive alternative for widespread adoption.
> > >
> > > > Re: Comparing Spark Transformer to Methods in Unstructured Pruning
> > >
> > > We understand the importance of comparing Spark Transformer with other related approaches. However, we would like to point out that different architectures were used as the base models in various papers, making direct performance comparisons less meaningful. Instead, the prevailing evaluation protocol, broadly adopted in works such as ReLUification, ProSparse, HiRE, and CATS, has been to compare the sparsified version of a model to its corresponding dense baseline and report the relative quality loss.
> > >
> > > In our submission, we follow this established practice by reporting the quality loss of Spark Gemma-2 relative to the vanilla (dense) Gemma-2, as seen in Table 2. Furthermore, during the rebuttal period, we added a comparison in the appendix (see Table 4), examining the amount of quality loss across different approaches relative to their respective dense counterparts. The results confirm that Spark Transformer incurs a quality loss of less than 1%, outperforming the majority of comparable methods.
> > >
> > > > Re: The alternative scenario where a pretrained model is given, the tradeoff between training time vs flops reduction and accuracy
> > >
> > > The reviewer raises an important point regarding the practicality of training a model from scratch versus fine-tuning a pretrained one. We agree that training from scratch involves higher initial costs. However, the long-term serving costs of models usually dominate overall expenditures, in which case the additional pre-training costs are amortized by the significant efficiency benefits realized from a higher level of sparsity and FLOPs reduction during model inference. Moreover, as hardware advances continue to favor sparsity, the trade-off further positions Spark Transformer as a viable choice for users and organizations prioritizing serving efficiency. We have added a discussion of these points to our manuscript and it will appear in the final version (as manuscript revision is no longer allowed at this point). We hope this expanded discussion clarifies our perspective and strengthens the case for Spark Transformer’s potential as a practical and impactful solution in the sparsity landscape.

---

> > > > ### Comment · Reviewer_uU7z · 2024-12-01
> > > >
> > > > Much of the concern from my response still remains unresolved.
> > > > One clarification request: The values provided in Table 4 don't seem to be an apples-to-apples comparison as they are gathered from each paper using different backbone and benchmarks, where a rough trend I see is that large models seem to suffer more from quality degradation. Can the authors provide an analysis of the presented data?

---

> > > > > ### Author Response · Authors · 2024-12-02
> > > > >
> > > > > We appreciate the reviewer’s insightful question about the impact of model scale on activation sparsity techniques.
> > > > >
> > > > > **Evidence from Table 4:**
> > > > > The data in Table 4 includes results from ProSparse applied to both LLaMA-7B and LLaMA-13B. The observed quality losses are 1.1% and 1.4%, respectively, suggesting a slight trend that larger models may experience more quality degradation. However, the difference between these two values is modest and may not be statistically significant enough to fully support this conclusion.
> > > > >
> > > > > **Broader Evidence:**
> > > > > Further insights on this topic can be found in related studies. The [Lazy Neuron Phenomenon](https://arxiv.org/abs/2210.06313) reports that larger T5 models exhibit increased activation sparsity (see their Fig. 2e-2f), while the [Sparsing Law](https://arxiv.org/abs/2411.02335v1) notes that activation ratios remain largely insensitive to model size across a series of decoder-only LMs (see their Fig. 7). These results suggest that larger models may not inherently be more challenging to sparsify.
> > > > >
> > > > > **Future Work:**
> > > > > We agree with the reviewer that analyzing the scaling behavior of activation sparsity techniques is an important area for further research. While our current work focuses on demonstrating sparsity effectiveness and preserving task quality, exploring scaling effects systematically is a valuable direction we aim to pursue in future studies.
> > > > >
> > > > > Once again, we thank the reviewer for their thoughtful feedback, which has enriched the discussion of this topic.

---

### Author Response · Authors · 2024-11-22

We thank the reviewer for their thoughtful, insightful, and constructive comments. This paper addresses the pressing need for an effective method to achieve a high level of activation sparsity in large language models (LLMs) without sacrificing quality. Our proposed Spark Transformer offers a simple architectural modification applicable to both FFN and Attention layers, achieving state-of-the-art FLOPs reduction while maintaining performance. We are grateful that Reviewers uU7z, 8pJV, and znbP recognize the importance of the problem we address, appreciate the novelty of our approach, and acknowledge the soundness of our mathematical explanations.

---

### Meta-Review · Area_Chair_u4dH · 2024-12-13

**Metareview:**

This paper produces an architectural variant of the transformer that uses sparse activations to reduce FLOP count. Reviewers recognised the importance of the problem tackled, and appreciated the method. The main concern of the reviewers was a lack of a demonstration of practical benefit, as there was no evidence that this theoretical reduction in FLOPs could lead to any real speed up on GPUs/TPUs.

This paper has very borderline scores (6,6,5,5) so I encouraged the reviewers to have a discussion to ideally come to a decisive opinion. All four reviewers then agreed on reject. The main reason cited was the lack of evidence of practical value (particular on GPUs), although one reviewer (despite agreeing on reject) doesn't believe this should be mainly attributed to this GPU evaluation given the merits of CPU acceleration for democratisation of LLMs. That said, there are issues of novelty, and a lack of fair comparisons.

I recommend reject; I do not see a good reason to overturn the decision of 4 reviewers. I would strongly suggest the authors take on board the suggestions of the reviewers, provide additional experiments (particular on GPUs), and submit to a future conference. Although this outcome will be disappointing, I believe the paper will become much stronger as a result of these exchanges.

**Additional Comments On Reviewer Discussion:**

There was some good discussion between the reviewers and the authors, with two reviewers raising their scores as a result (3->5, 5->6). My final decision was mainly based on the final discussion between the reviewers when they reached a consensus. The main concern of there not being a GPU implementation wasn't explicitly addressed by the authors, so went unresolved (although the authors believe this should be possible, and potentially work well).

---

### Decision · Program_Chairs · 2025-01-22

Reject